# Exploring Spatiotemporal Pattern of Grassland Cover in Western China from 1661 to 1996

**DOI:** 10.3390/ijerph16173160

**Published:** 2019-08-29

**Authors:** Fan Yang, Fanneng He, Shicheng Li, Meijiao Li

**Affiliations:** 1Key Laboratory of Land Surface Pattern and Simulation, Institute of Geographic Sciences and Natural Resources Research, Chinese Academy of Sciences, Beijing 100101, China; 2University of Chinese Academy of Sciences, Beijing 100049, China; 3School of Public Administration, China University of Geosciences, Wuhan 430074, China

**Keywords:** grasslands, reconstruction, land use and land cover, land reclamation, immigration, western China

## Abstract

Historical grassland cover change is vital for global and regional environmental change modeling; however, in China, estimates of this are rare, and therefore, we propose a method to reconstruct grassland cover over the past 300 years. By synthesizing remote sensing-derived Chinese land use and land cover change (LULCC) data (1980–2015) and potential natural vegetation data simulated by the relationship between vegetation and environment, we first determined the potential extent of natural grassland vegetation (PENG) in the absence of human activities. Then we reconstructed grassland cover across western China between 1661 and 1996 at 10 km resolution by overlaying the Chinese historical cropland dataset (CHCD) over the PENG. As this land cover type has been significantly influenced by anthropogenic factors, the data show that the proportion of grassland in western China continuously decreased from 304.84 × 10^6^ ha in 1661 to 277.69 × 10^6^ ha in 1996. This reduction can be divided into four phases, comprising a rapid decrease between 1661 and 1724, a slow decrease between 1724 and 1873, a sharp decrease between 1873 and 1980, and a gradual increase since 1980. These reductions correspond to annual loss rates of 7.32 × 10^4^ ha, 2.90 × 10^4^ ha, 17.04 × 10^4^ ha, and −2.37 × 10^4^ ha, respectively. The data reconstructed here show that the decrease in grassland area between 1661 and 1724 was mainly limited to the Gan-Ning region (Gansu and Ningxia) and was driven by the early agricultural development policies of the Qing Dynasty. Grassland was extensively cultivated in northeastern China (Heilongjiang, Jilin, and Liaoning) and in the Xinjiang region between 1724 and 1980, a process which resulted from an exponential increase in immigrants to these provinces. The reconstruction results enable provide crucial data that can be used for modeling long-term climate change and carbon emissions.

## 1. Introduction

Human activities over long time periods have significantly altered the landscape of the Earth’s surface by transforming natural ecosystems to agricultural areas [1,2]. These modifications have led to serious degradation of terrestrial ecosystems and have continuously influenced climate change and human sustainable development [3,4,5,6]. In light of the rapid development and application of remote sensing (RS)-derived datasets and advanced GIS techniques, land use and land cover change (LULCC) datasets obtained from satellite images have the potential to constrain these spatial patterns and make a contribution to our understanding of past land use changes. Indeed, historical LULCC studies have gradually moved past their previous limitations (i.e., the results have been qualitative, statistically insignificant); new spatially explicit reconstruction results have been widely utilized in modeling climate change, carbon emissions, and the ecological environmental effects due to human activities [7,8,9,10]. Numerous scholars and research projects therefore list the reconstruction of historical LULCC and an understanding of its effects as one of their core aims [11,12,13].

Considerable progress has been made towards understanding historical LULCC. Long-term cropland and pasture coverage estimates are the main objects of several well-known global land use datasets, including global croplands and pastures—SAGE (Center for Sustainability and the Global Environment) [14,15], anthropogenic land use estimates for the Holocene—HYDE (History Database of the Global Environment) [16,17,18], global agricultural areas and land cover—PJ (Pongratz Julia) [19], and past anthropogenic land cover change scenario—KK10 (Kaplan and Krumhardt 2010) datasets [20,21]. Grassland is one of the most widespread land covers on Earth, covering nearly 35 million km^2^ of the world’s land surface [22]. Nevertheless, earlier work in this area has mainly concentrated on agricultural land (i.e., croplands and pastures), and a limited number of studies have so far aimed to reconstruct historical grassland cover. In some examples, researchers used RS-derived datasets and historical maps as sources to reconstruct historical grassland cover at the regional scale across the eastern United States between 1650 and 1992 [23], across southeast Sweden between 1700 and 2000 [24], across central Europe between 1900 and 2010 [25]. Estimates for grassland proportions in different global political units between 1700 and 1992 were provided in the initial version of the SAGE dataset by overlaying cropland cover and potential natural vegetation (PNV) [15]. Similarly, a figure for global millennium natural grassland was reconstructed in the PJ dataset by subtracting agricultural areas from PNV [19].

For China, grasslands comprise the largest terrestrial ecosystem, accounting for approximately 41% of the national land area [26,27,28]. On the one hand, grasslands provide forage for animal production. On the other, as Chinese grassland is mainly distributed in areas characterized by fragile ecological environments, significant topographic fluctuations, or both, this land cover also plays an important role in wind-breaking and sand fixation, as well as in water, soil, and biodiversity conservation [22]. Grassland resources are therefore of significant economic value and ecological function. Changes in this kind of land cover have also precipitated a series of past climatic and ecological environmental problems. Accelerated soil erosion, frequent sandstorms, and the development of vast saline lands in northern China since the end of the 20th century have been attributed to grassland shrinkage and degradation [28]. It is, therefore, necessary to know the distributions, magnitudes and dynamic changes over time of grassland in China.

Although global datasets such as PJ and SAGE contain long-term grassland data in China, estimates for the grasslands in them vary widely. Previous studies have evaluated Chinese grassland data within global datasets using sporadic regional studies [29,30]. Results indicate that global datasets have so far failed to reflect spatiotemporal changes in Chinese grasslands. Large differences can be seen between the global datasets (i.e., SAGE and PJ) and regional research on cropland quantification [31,32]; uncertainties from this land use type have been propagated into grassland cover reconstructions, because the most commonly utilized method deducts agricultural land from PNV. Such global data have no contribute to understanding regional dynamic changes of grassland resources in this region.

A few scholars have carried out projects in this area over recent decades aimed at reconstructing Chinese historical grasslands. In one example, Ge et al. [33] calibrated various kinds of LULCC data to reveal the processes underlying shrinking grassland areas across China between 1935 and 1997, while Ye et al. [34] and Wu et al. [35] estimated these areas across northeastern China and within the Huangshui River valley in Qinghai Province over the last 300 years by analyzing historical documents and recovering potential natural grasslands, respectively. Compared with progresses on reconstructing historical croplands [13,36,37,38,39] and forests in China [40,41,42], historical grassland reconstructions have been limited to just single periods or regions where abundant historical documents are found. This situation has seriously constrained our understanding of historical grassland changes and their effects on the eco-environment.

In view of the uncertainties of Chinese grassland in global datasets and the sporadic regional grassland reconstructions in China, as well as the fact that western China is the major sink of current national grassland resources (approximately 80% of total Chinese grassland area), this study therefore aims to reconstruct historical grassland cover across western China over the past 300 years. Specifically, building on an understanding of the characteristics and driving forces of grassland change in western China, we propose a method for reconstructing historical grassland cover across this region and further map grassland cover between 1661 and 1996.

## 2. Materials and Methods

### 2.1. Study Area

This study emphasizes the extent of grassland cover across western China [43]. Data show that grassland at the provincial level is mainly distributed within Xinjiang, Tibet, Qinghai, Gan-Ning and Inner Mongolia, while important additional regions include the western part of northeastern China (Heilongjiang, Jilin, and Liaoning), the northern part of Jing-Jin-Ji, Shanxi and Shaanxi, the western part of Chuan-Yu, and the northern part of Yunnan. This current grassland distributional range involves 13 provinces which cover an area of 759.26 × 10^6^ ha and encompass a range of landforms including plateaus and basins (Figure 1). Among these, the Qinghai-Tibet Plateau, the Third Pole of the world, comprises extremely high mountains, high mountains, and plateaus, while the second kind of terrain in China includes the Tarim, Dzungaria, and Sichuan basins, as well as the Inner Mongolian, Loess, and Yunnan-Guizhou plateaus. The main climatic regimes within this region include an alpine zone on the Qinghai-Tibet Plateau, typical semi-arid and arid continental zones in northwest China, and a subtropical monsoonal zone in the southwest. Average annual precipitation across the whole region is less than 600 mm (excluding Chuan-Yu and Yunnan). Historically, compared with eastern China, this study area is sparsely populated and rich in grassland resources. The natural conditions are conducive to the development of animal husbandry, and so western China has always been a traditional pastoral region and a major distribution center for nomads.

China is a traditional agricultural country and has long been characterized by large-scale cropland agriculture and small-scale animal husbandry, despite abundant grassland resources in the west [29]. Indeed, over the last 2000 years, western China has experienced four stages of large-scale immigration, including during the Qin-Han, Sui-Tang and the mid-Qing dynasties, as well as throughout the mid-20th century [44,45]. In the course of safeguarding the security of borderlands in western China, grassland was often regarded as an important agricultural reserve resource by successive dynastic governments. As the agricultural population has migrated from the mid-east into western China, agriculture has intensified and the range of land used for this function has expanded, resulting in the large-scale conversion of grasslands to cultivated land.

To facilitate analyses of grassland cover change at the provincial scale, provincial boundaries were used here, although ranges for this land cover type are inconsistent with these administrative divisions. The modern Chinese national boundary was therefore fixed on the base map for this study, and some provinces were regrouped due to frequent changes in national territory and provincial units over the last 300 years. Beijing, Tianjin, and Hebei were merged into the Jing-Jin-Ji area, while Gansu and Ningxia were merged into Gan-Ning area and Sichuan and Chongqing were merged into the Chuan-Yu area [41]. Moreover, this study adopts the grassland definition as used in phytogeography, and thus includes all natural vegetation types that are dominated by herbaceous plants, as well as all communities of this type other than crops [46].

### 2.2. Data Sources

The historical cropland data used here were derived from the Chinese historical cropland dataset (CHCD) presented by Li et al. [37]. The spatial resolution of the CHCD is 10 km. In earlier work, Ge et al. [47] were the first to estimate Chinese provincial cropland area over the last 300 years using historical documents, population data, and government inventories. Afterwards, Li et al. [37] assessed the suitability of land for cultivation by quantifying factors related to cropland spatial distributions, including altitude, surface slope, and climatic potential productivity, and further utilized weights based on the degree of land suitability for cultivation to allocate cropland to 10 km resolution grids [37]. Uncertainty analysis showed that the reconstruction results can objectively reveal the spatiotemporal changes of historical cropland. The CHCD is available for the years 1661, 1724, 1784, 1820, 1873, 1933, 1980, and 1996.

The PNV map developed by Ramankutty and Foley [15] was used in this analysis. This data was synthesized from the DISCover land cover dataset and the modeled natural vegetation. This vegetation map comprises 15 types at 5 min resolution. Results show that the PNV spatial pattern for northeastern China is similar to the macro-scale native vegetation pattern reconstructed by Zhang et al. [48], and the difference of grassland coverage is small between them; that is, the potential grassland is 4.2% lower than the native grassland (Table 1). It suggests that this PNV data is reliable and is therefore used here to replace the regions dominated by anthropogenic land use in western China.

A series of RS-derived land use/cover data of western China encompassing the period between 1980 and 2015 were obtained from Chinese LULCC datasets (CLUDs, available on http://www.resdc.cn). Liu et al. [49,50] used the human–computer interactive interpretation method, as well as Landsat TM/ETM digital images covering China to construct CLUDs. The land use/cover types in CLUDs include six classes—cropland, forestland, grassland, water body, unused land, and built-up land—along with 25 subclasses. These CLUDs, with a resolution of 1 km, were updated regularly at 5-year intervals from 1980 to 2015, and the quality of the datasets was ensured through uniform quality control and integration checking. To assess the accuracy of the database, a large number of field-investigation records were obtained. Results indicate the accuracy of the six classes of land use/cover type was greater than 94.3%, and the overall accuracy of the 25 subclasses was greater than 91.2%, which satisfies the demands of user mapping accuracy at a scale of 1:1 million.

### 2.3. Methodology

The process of anthropogenic land use in western China suggests that population growth has accelerated the conversion of natural grasslands into croplands. It is, therefore, clear that changes in grassland cover have been significantly influenced historically by land reclamation. This means that available and reliable historical cropland data across this region can be used to reflect reclamation of grassland because these data are the quantitative expression of land reclamation behavior.

Based on this knowledge, we propose a method for reconstructing historical grassland cover (Figure 2). This involves two steps: (1) determining the potential extent of natural grassland (PENG) in the absence of anthropogenic activities; and (2) building a grassland cover reconstruction model by combining the PENG with the CHCD.

#### 2.3.1. The PENG in the Absence of Human Land Use

For the application using cropland data to extrapolate backward grassland cover change, it is essential to know the land surface properties of western China before agriculture emerged. Native vegetation reconstructed based on historical evidence (e.g., historical archives, pollen, or archaeological observations) can objectively reflect the land cover before land reclamation [48]. However, the reconstruction results only provide a macro-spatial pattern [51], which cannot meet the demands of exploring land cover change on a large spatial scale. At this case, scholars generally replace native vegetation with PNV as a starting point for analyzing land cover change [15,19].

The main reconstruction methods for PNV are as follows: (1) extrapolation of existing natural vegetation to similar habitats (a simple method of qualitative estimation) [52]; (2) simulating PNV by using the relationship between vegetation and environment [53,54]; (3) synthesizing (1) and (2) to determine PNV. The third method is an optimization of the previous methods and is currently widely used in the historical LULCC reconstruction field [15,19,41]. Therefore, we adopt the method to determine PNV and make necessary adjustments and supplements according to the characteristics of western China.

Indeed, Lan [55] pointed out that small parts of the modern grassland area might also be forested in the upper Yangtze River historically, and the current grassland vegetation is the result of the interaction of human activities and natural environment. Based on the third method and the above findings, it is appropriate to assume that most modern grassland regions detected using RS across western China were also grassland regions historically and, indeed, small parts might also be forested; modern non-grassland regions monitored by RS (i.e., those dominated by anthropogenic land use, including cultivated regions and built-up areas) might also have been grasslands in the past and depends on the PNV distribution.

Specifically, we extracted the distributions of modern natural vegetation (forest, grassland) and anthropogenic land cover (cultivated regions and built-up areas) in western China from CLUDs, while potential grassland and forest areas were obtained from the PNV data by Ramankutty and Foley [15]. After grid cells of potential forest were subtracted from modern grassland cover, the remainder were classified as both the past and present grassland distributional area. Superimposing potential grassland cover over that seen to currently be non-grassland cover, the overlapped grid cells were classified as historical natural grassland cover in the non-grassland regions. Finally, combining these maps for modern grassland and the non-grassland regions enabled the determination of the PENG in the absence of land reclamation.

#### 2.3.2. Reconstructing Grassland Cover

Population growth has long been considered the main driver of LULCC [21,56]. As population increases in a given region, so does demand for food. Croplands therefore need to provide more food. Because of the backward productive level in historical times, the increase of grain yields was mainly due to the expansion of cropland [19,21]. For western China, the increase of cropland has encroached on a large amount of grassland cover. In addition, the cropland in this region is mainly distributed around urban built-up areas and rural settlements both in the past and at present. As the amount of cropland increases along with population growth, residential land area also expands. Statistical results of CLUDs show that grassland coverage decreased as cropland area expanded; as the proportion of cropland was greater than, or equal to 90%, corresponding grassland coverage was mostly less than 1% because the remainder was usually occupied by residential land (Figure 3).

Modern LULCC information can be obtained through the monitoring of satellite remote sensing, aerial photographs, and field surveys; past LULCC is, however, limited in space and time due to the lack of direct, large-scale observations [57]. Therefore, the common method used by historical LULCC reconstruction scholars is to set some reasonable assumptions and then propose a past LULCC scenario based on the relevant knowledge of modern LULCC [15,18,19,20,21]. Based on the above analysis, we believe that the correlation between modern cropland and grassland shown in Figure 3 is also applicable to the past. This study therefore assumed that historical grassland coverage was zero in a grid if the proportion of cropland was greater than 90% in western China.

Grassland cover for different years was therefore reconstructed on the basis of these assumptions by overlaying CHCD over the PENG (Equation (1)). On this basis, the amount of grassland across the whole study region as well as in provinces over the last 300 years was reconstructed using spatial aggregation to reveal regional and temporal variations in grassland resources.

This relationship was expressed as follows:(1)$$G(j,t)\;=\alpha \;\times \;[\overset{100}{\underset{i=1}{\sum }}\mathrm{PG}(i,j,t)\;-\;C(j,t)]$$In this expression, G (j, t) denotes grassland area in grid j (10 km) in year t, while PG (i, j, t) refers to the ith PENG grid (1 km) in grid j without land reclamation in year t, C (j, t) denotes the jth CHCD grid (10 km) in year t, and α is the ratio between grassland and cultivated land. When C (j, t) is less than 90 km^2^, α is 1 otherwise the value is 0.

## 3. Results

### 3.1. Changes in Grassland Area Across Western China

Changes in grassland area across western China between 1661 and 1996 are illustrated in Figure 4. Data show that reconstructed grassland areas across this region in 1661, 1724, 1784, 1820, 1873, 1933, 1980, and 1996 were 304.48 × 10^6^ ha, 299.87 × 10^6^ ha, 298.46 × 10^6^ ha, 296.99 × 10^6^ ha, 295.54 × 10^6^ ha, 288.09 × 10^6^ ha, 277.31 × 10^6^ ha, and 277.69 × 10^6^ ha, respectively. Over the time period between 1661 and 1996, a total grassland area of 26.79 × 10^6^ ha was lost in western China at an annual loss rate (ALR) of 8.00 × 10^4^ ha. Results suggest that changes in grassland over this time period passed through four phases due to changes in the intensity of land reclamation, between 1661 and 1724 (P1), between 1724 and 1873 (P2), between 1873 and 1980 (P3), and between 1980 and 1996 (P4).

Grassland area decreased rapidly during P1. This is likely because the Qing government vigorously developed agriculture throughout the surrounding areas of the Xinjiang Province to feed the army and quell the Junggar uprising [58,59]. This resulted in a decrease in grassland area from 304.48 × 10^6^ ha in 1661 to 299.87 × 10^6^ ha in 1724; this translates to a loss of 4.61 × 10^6^ ha, 16.97% of the total loss, an ALR of 7.32 × 10^4^ ha over an almost 60-year period.

Once Xinjiang Province had been formally amalgamated into the territories of the Qing Dynasty and there was a rapid decrease in demand for military food supplies, decreases in grassland area remained insignificant throughout P2 [60,61], in part also due to limited population levels in western China [62]. In 1873, grassland area was 295.54 × 10^6^ ha; this means that 4.33 × 10^6^ ha of grassland in total was lost over this 150-year period, just 15.92% of the total loss and similar to P1. At the same time, however, the ALR over this period was just 2.90 × 10^4^ ha, less than half that seen during P1.

The sharpest decrease in the grassland area was seen during P3 as a large number of immigrants moved into this border area since the mid-19th century. This led to a significant transformation in the agricultural economy from military-oriented croplands developed by the Qing government to agricultural exploitation due to spontaneous immigration [63]. Grassland area was 277.31 × 10^6^ ha in 1980 implying a loss of 18.23 × 10^6^ ha over this more than 100-year period, four times greater than during either P1 or P2. This accounts for 67.11% of the total loss, an ALR of 17.04 × 10^4^ ha.

Following 1980 (P4), after a series of grassland protection policies were introduced by the government, this area began to gradually increase; by 1996, grassland area had increased to 277.69 × 10^6^ ha and increments had reached 0.38 × 10^6^ ha over an almost 20-year period, an ALR of −2.37 × 10^4^ ha.

### 3.2. Changes in Provincial Grassland Areas

Grassland area change magnitudes varied significantly at the provincial level (Figure 5). Data suggest that the Loess Plateau region (i.e., Shanxi, Shaanxi, and Gan-Ning), as well as the Jing-Jin-Ji and Southwest China (Chuan-Yu and Yunnan), all exhibited the same trends, as seen across the whole of western China, including a rapid decrease during P1, a slight decrease during P2, a dramatic decrease during P3, and a gradual increase in P4. Throughout the whole period, for instance, ALRs across Shanxi Province were 0.18 × 10^4^ ha, 0.10 × 10^4^ ha, 0.21 × 10^4^ ha, and −0.11 × 10^4^ ha, respectively, while those for the Chuan-Yu region were 0.33 × 10^4^ ha, 0.15 × 10^4^ ha, 1.38 × 10^4^ ha, and −0.12 × 10^4^ ha, respectively.

Changes in grassland area in some provinces deviated from those seen across western China as the intensity of clearing increased continuously between P1 and P3. In particular, a large proportion of grassland area was converted into cropland throughout the late 19th century because of substantial immigration. These regions included northeastern China (i.e., Heilongjiang, Jilin and Liaoning), Inner Mongolia, Xinjiang, and on the Qinghai-Tibet Plateau (i.e., Tibet and Qinghai). Indeed, for example, ALRs for Jilin Province between P1 and P4 were 0.04 × 10^4^ ha, 0.66 × 10^4^ ha, 2.25 × 10^4^ ha, and −0.45 × 10^4^ ha, respectively, while those for Xinjiang Province were 0.10 × 10^4^ ha, 0.16 × 10^4^ ha, 3.16 × 10^4^ ha, and −0.39 × 10^4^ ha, respectively.

### 3.3. Grassland Cover Spatial Patterns

The spatially explicit grassland cover is shown in Figure 6. These reconstructions show that the area of PENG accounted for 320.72 × 10^6^ ha in the absence of human activities (Figure 6a) and that the area of grassland cover continuously decreased over time because of land reclamation activities (Figure 6b–j).

To better understand spatiotemporal variations in grassland cover across western China, net changes in this land cover type for the periods between 1661 and 1724, 1724, and 1873, 1873 and 1980, and 1980 and 1996 were calculated by subtracting values for the first years of each period (i.e., 1661, 1724, 1873, and 1980, respectively) from the latter (i.e., 1724, 1873, 1980, and 1996, respectively; Figure 7). Over the past 300 years, clearing of grassland mainly occurred in the western part of northeastern China, the northern margin of Tianshan Mountains, the surrounding area of Tarim basin and the central and eastern part of Gan-Ning.

## 4. Discussion

### 4.1. Forces Driving Grassland Changes

Demographic, socioeconomic, and policy considerations have all played important roles in grassland resource dynamics. Throughout the early Qing Dynasty period (P1), reclamation of grassland was mainly limited to the Gan-Ning region where was the military supplies rear for quelling the Junggar uprising (Figure 7a). To ensure provisions for the army and to consolidate the rear of this campaign, the government encouraged farmers to open up wastelands by reducing taxes [58,59]. This policy was enacted by the emperors Kangxi and Yongzheng (1661–1735) [59]. Enthusiasm for land reclamation was greatly enhanced at that time, which resulted in the large-scale conversion of grasslands into croplands.

The agricultural frontier gradually migrated northwards into Inner Mongolia and northeastwards into northeastern China throughout the middle period of the Qing Dynasty (P2) [64], but agricultural intensification remained at a low level (Figure 7b). Throughout this period, in order to protect Northeast China (the birthplace of the Qing Dynasty), the government implemented a policy of prohibiting reclamation from the reign of Emperor Kangxi onwards and so the ‘Wicker Frontier Wall’ became a dividing line between agriculture and animal husbandry [64]. It also remained impossible throughout this period to completely prevent refugees from entering this region from the surrounding close provinces (e.g., Shandong, Jing-Jin-Ji, Shanxi, and Henan). As a result, the enlargement of privately reclaimed land transformed large areas of grassland into cropland.

Throughout the late Qing Dynasty period, conflicts between rapid population increases in the traditionally cultivated regions of China and limitations in such land resources coupled with natural disasters and chaos caused by wars became major societal problems. Indeed, at that time, approximately 80% of the Chinese population lived in eastern agricultural areas but as the number of people grew rapidly, limited cropland resources struggled to sustain these levels [44]. Generally, per capita cropland area (PCA) is often used to weigh food security in a country or region [65] and this variable can be calculated by dividing cropland area by human population data. The minimum PCA refers to the area necessary to support a person; previous studies suggest that minimum PCA was 0.27 ha in the north of China under productive conditions, given the technological level at that time [66,67]. Data suggest that since the 19th century, the PCA of northern China was around, or below, the minimum threshold, including Jing-Jin-Ji, Shanxi, Shandong, and Henan (Figure 8). A bumper year was therefore barely sufficient to support the livelihoods of thousands of farmers; this meant that the agricultural economy tended to collapse regularly, while the number of refugees increased geometrically in famine years. Population increase in this traditional agricultural region and the resultant agricultural economic crisis were therefore the main causes of a large number of refugees.

Against this background of demographic and socioeconomic pressure, the Qing government was forced to abolish its ‘Prohibit reclamation in Northeast China’ policy and encourage immigration into borderlands [60,64]. It is noteworthy that the population of northeastern China was less than 1.00 million people in 1776 and had increased to 41.73 million people by 1953 (41.24 times). At the same time, the population of Xinjiang increased from 0.86 million people in 1880 to 4.76 million in 1953 (5.52 times) (Figure 9) [62,68]. Losses in grassland were closely associated with population growth and had determination coefficients (R^2^) of 0.86 in northeastern China and 0.91 in Xinjiang, respectively. Because of this influx of immigrants, northeastern China and Xinjiang experienced serious breakdowns in grassland resources from the time of the late Qing Dynasty to the 1980s (P3) (Figure 7c). Spontaneous immigration eventually thoroughly altered the nomadic economy of this region into an agricultural one [63].

The area of grassland gradually increased in western China subsequent to the 1980s (P4) (Figure 7d). Thus, when the first national Grassland Law was initiated in 1985, these regions in western China had entered a new constant expansion developmental stage alongside the implementation of a series of protection policies, which included prohibiting reclamation, enforcing proper grazing, and replacing croplands with pastures.

### 4.2. Differences between Global Datasets and the Results of This Analysis

The process of reconstructing historical LULCC involves clarifying land use practices in a given region, selecting corresponding concepts, devising an appropriate reconstruction method, and then reconstructing relevant data. Huge differences are nevertheless seen in the selection of definitions and reconstruction mechanics due to the transformation and utilization of land cover in a country or region that is deeply influenced by regional environmental conditions (e.g., climate, altitude, topography) and local culture (e.g., dietary structures) [29,69].

For Europe and the United States, there has been such a high proportion of animal husbandry and a large amount of pasture over a long period of time especially in the Alps of central Europe and the western mountain areas of the United States [70]. Land use strategy in these regions is that population growth stimulates the development of animal husbandry and therefore enlarges pasture areas [18,19,20]. The well-known HYDE dataset applies this Europe and United States land use strategy on a global scale and adopts the use of the term ‘pasture’ as used in the agronomy and resource sciences, a general term for all types of grazing land including natural and anthropogenic grasslands, deserts, shrubs, and sparse forests [71,72,73]. Therefore, global pasture area is estimated by multiplying per capita pasture with population data. It is important to remember, however, that this currently applied definition for pasture is imprecise and lacks any characterization of animal production systems on studies of LULCC [74]. This indefinite content results in varying inclusions and exclusions when global datasets and statistical inventories are compared [70,74].

In China, in contrast, a dense population, the synchronization of rain and heat, and developed traditional agriculture characterized the eastern regions of the country over long periods of time, while the west remained sparsely populated and characterized by small-scale animal husbandry. The primary trend in Chinese historical grassland change has therefore been a decrease in natural areas affected by land reclamation rather than increases in pasture [33]. The last 300 years have witnessed the continuous expansion of agriculture into pastoral areas and land use practice means that more grassland is likely to be converted into cropland as population grows. Based on typical land use practice across western China, this study adopts the grassland definition as used in phytogeography [46]. Available data on land reclamation were then also utilized to capture the history of grassland changes.

The HYDE dataset implies that a large proportion of natural grasslands and forests have been converted into pastures to meet human demands in many regions of the world. This dataset also assumes that major land use practices in China are the same as those in Europe and the United States, comprising large-scale increases in herbaceous planting vegetation. The graph in Figure 10 shows that the HYDE 3.2 reconstruction for pasture in western China is characterized by dramatic growth, an increase from 47.76 × 10^6^ ha in 1700 to 355.47 × 10^6^ ha in 2000 (approximately 7.4 times). In contrast, existing historical data and our current understanding do not support this relationship; the reconstruction presented here implied a gradual decrease over the last 300 years. Therefore, land use practices, the definition of pasture, and the reconstruction method used by the HYDE 3.2 dataset are not applicable to China.

Pasture area was also estimated in the PJ dataset using a similar method to that employed by the HYDE 3.2 dataset. Natural grassland area was reconstructed by deducting agricultural zones (including cropland and pasture) from potential grassland vegetation [19]. This enabled calculation of the total area of herbaceous vegetation across western China via the sum of pasture and natural grasslands from the PJ dataset. This PJ dataset result suggests that the area of herbaceous vegetation in western China increased from 219.81 × 10^6^ ha in 1700 to 274.16 × 10^6^ ha in 1980 (Figure 10). This result also seems to suggest that the pasture increment is larger than that of natural grassland transformed into other types of land use, including croplands and built-up areas. Large discrepancies nevertheless exist between the PJ dataset and the reconstruction presented here. It is clear, therefore, that global datasets like HYDE 3.2 and PJ are unable to objectively reveal the herbaceous vegetation dynamics of western China.

Currently, numerous regional grassland cover has been reconstructed outside China [24,25,75,76]. Generally, the research thoughts of different regions are consistent, that is, a large amount of grassland was converted into cropland and pasture due to population growth. However, there are obvious regional differences in reconstruction methods because of different anthropogenic land use practices. It is, therefore, not feasible to reconstruct historical grassland cover with a uniform method on global scale; conversely, the reconstruction results in this way are uncertain at regional scale, such as HYDE 3.2 and PJ. The idea of dividing different regions, time sections and fine categories can be adopted to reconstruct spatial pattern of historical grassland.

### 4.3. Limitations

Determination of PENG in this analysis was carried out by synthesizing RS-derived LULCC data and PNV data derived from modeling. Some previous scholars have tried to restore native grassland vegetation using historical evidence (e.g., historical archives, pollen, or archaeological observations) and results to date show that use of the latter provides more accurate results. Reconstruction results that are based on native grassland vegetation can more accurately describe dynamic changes; however, the lack of richness in proxy data to date means that applications are limited at the local scale. Native grassland vegetation across the whole study area will enable more precise data and can contribute to better depictions of the history of this land cover type across western China.

One better solution for estimating grassland at present would be to combine CHCD with PENG, given the fact that changes in the former have mainly been forced by land reclamation across western China. Historically, other than large-scale land reclamations, changes in grassland cover have also been influenced by numerous natural and cultural factors including climate change, destruction by wars, and the development of built-up areas. It remains difficult to quantify these issues, however, because the historical record is highly fragmented spatiotemporally. These factors were therefore deliberately omitted from this analysis.

The natural landscapes of China (including natural forests and grasslands) have undergone dramatic historical changes, which have exerted profound impacts on regional climate change. A range of historical data is currently available to reconstruct cropland distributions in the past; rich Chinese historical records at least support the fact that croplands have been reconstructed over the last millennium [38]. Records for forests and grasslands remain rare and qualitative, however; reconstructing this land cover type therefore remains a huge challenge. A lack of historical data globally also means that the methods available to reconstruct grassland areas in the past remain immature and are still at an exploratory stage. As results are not yet accurate in this area, it remains important to develop new approaches to objectively and reliably reflect change trends. The results of this analysis are a step in the right direction.

Despite limits to historical data, grassland types across western China mainly comprise zonal grassland vegetation, and changes in this land cover type have been affected by land reclamation. This means that reconstructing grassland can be done if certain rules are followed (i.e., land reclamation is the dominant force); this is also the main reason this land cover type can be reconstructed by this study. In eastern and southern China, grassland vegetation is dominated by non-zonal secondary growth, which is the result of zonal forest vegetation and succession to secondary grassland because of anthropogenic activities. Irregular changes in grassland and the lack of relevant historical records for this region present a series of very significant challenges for grassland reconstructions in China.

### 4.4. The Use of Historical Grassland Data

LULCC profoundly affects regional and global climate change via a range of biogeophysical and biogeochemical mechanisms [77]. Carbon emissions over recent decades have received particular attention around the world [2,20]. In this context, Houghton and Hackler [78] were the first to estimate carbon emissions from human land use in China over the past 300 years. The results are not reliable, however, because of uncertainties in the LULCC data they used; this work was subsequently improved by Ge et al. [79], who re-worked estimates of carbon emissions from land use over the last 300 years based on cropland and forest reconstruction data produced in China. Although grasslands comprise the largest terrestrial ecosystem within China, this land cover type is not taken into account in estimates because changes are not well documented. This issue suggests that the carbon emissions calculated by Ge et al. [79] might be markedly lower than the actual values, although real emissions remain unknown. The historic grassland cover data presented here will therefore contribute to solving this problem and will augment our understanding of human land use-induced carbon emissions.

## 5. Conclusions

Historical grassland reconstruction is crucial for modeling long-term climate change and carbon emissions. Based on land use practice in western China, this study presents a reconstruction model for historical grassland cover using PENG and CHCD. In addition, grassland cover maps of western China were created at a 10 km resolution for 1661–1996. Results suggest that the natural grassland area across western China (without land reclamation) was 320.72 × 10^6^ ha; and area decreasing trends were observed in all western provinces for 1661–1996.

Grassland cultivation mainly occurred in the Gan-Ning area and was driven by agricultural policies developed by government of the Qing dynasty between 1661 and 1724. Agriculture subsequently expanded in Inner Mongolia and northeastern China due to an increased number of refugees; this led to a slight decline in grassland area for 1724–1873. Over the period between 1873 and 1980, cultivation of grasslands was mainly concentrated in northeastern China and Xinjiang. Large increases in both the intensity and extent of reclaimed grassland for this period resulted from agricultural economic crisis, policies, and exponentially increased immigration.

Definitions for pasture, land use practice, and the reconstruction methods applied by global datasets, including HYDE 3.2 and PJ, are not suitable for western China. The global datasets fail to objectively reveal the influence of long-term anthropogenic activities on grassland cover of western China.

## Figures and Tables

**Figure 1 ijerph-16-03160-f001:**
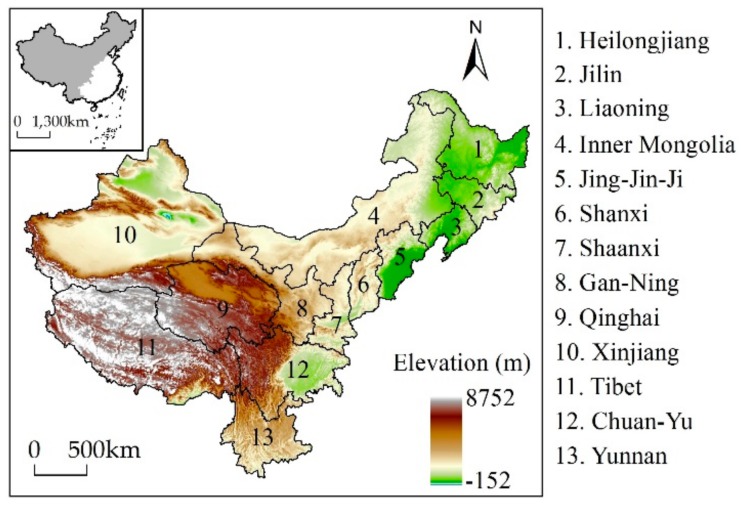
Maps showing the location and topography of grassland across western China. The grassland region encompasses 13 provinces. The digital elevation model (DEM) was obtained from Geospatial Data Cloud (http://www.gscloud.cn/).

**Figure 2 ijerph-16-03160-f002:**
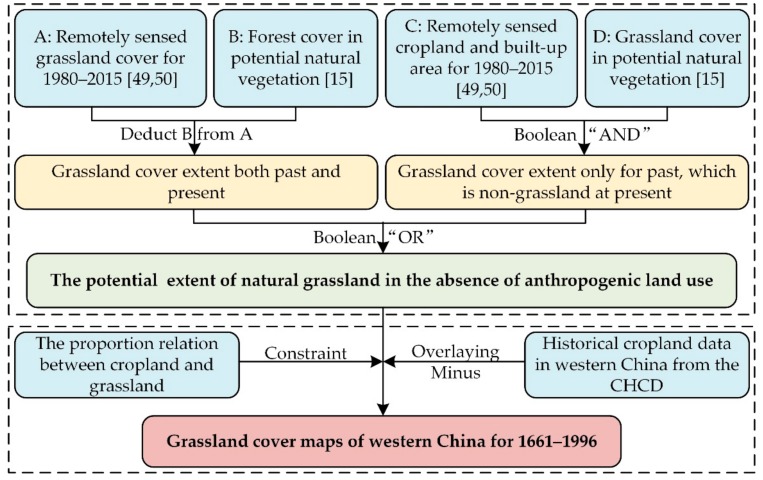
Proposed scheme for reconstructing historical grassland cover across western China. Abbreviation: CHCD, Chinese historical cropland dataset for the past 300 years.

**Figure 3 ijerph-16-03160-f003:**
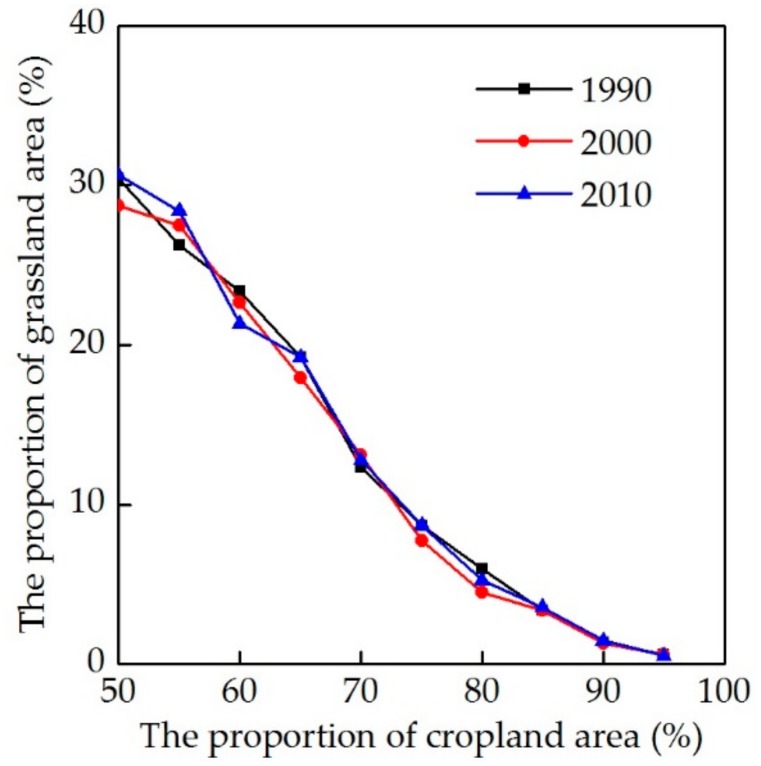
Proportional relationship between modern cropland and grassland area.

**Figure 4 ijerph-16-03160-f004:**
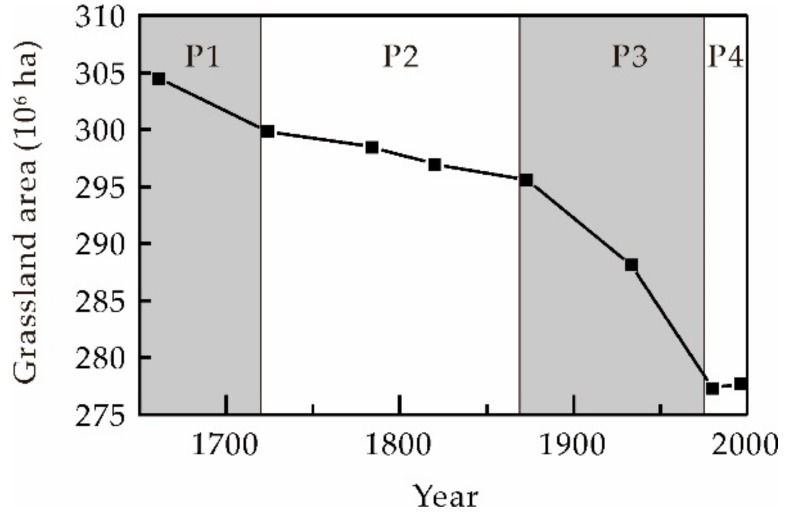
Total grassland area across western China between 1661 and 1996.

**Figure 5 ijerph-16-03160-f005:**
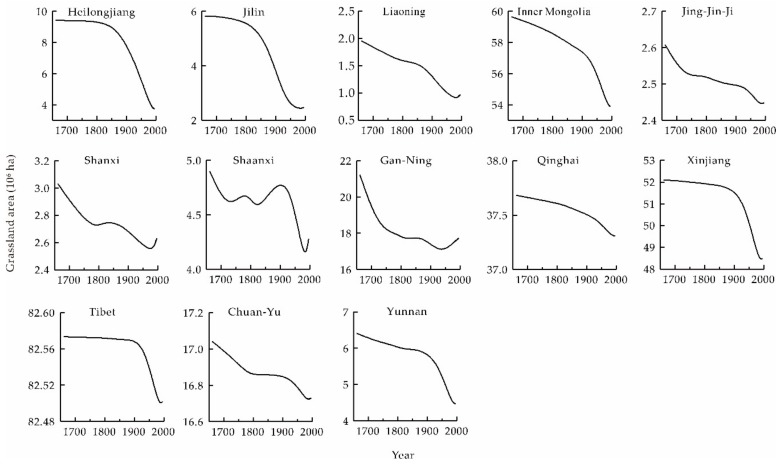
Changes in provincial grassland area across western China between 1661 and 1996. Data curves were smoothed using B-spline function.

**Figure 6 ijerph-16-03160-f006:**
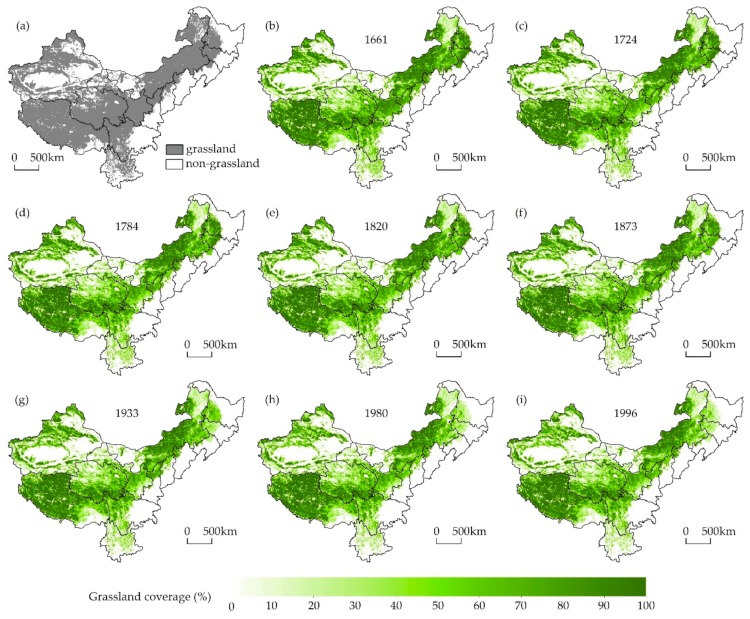
Maps showing grassland cover across western China for (**a**) the period without land reclamation; (**b**) 1661; (**c**) 1724; (**d**) 1784; (**e**) 1820; (**f**) 1873; (**g**) 1933; (**h**) 1980; and (**i**) 1996. No grasslands are present within white areas.

**Figure 7 ijerph-16-03160-f007:**
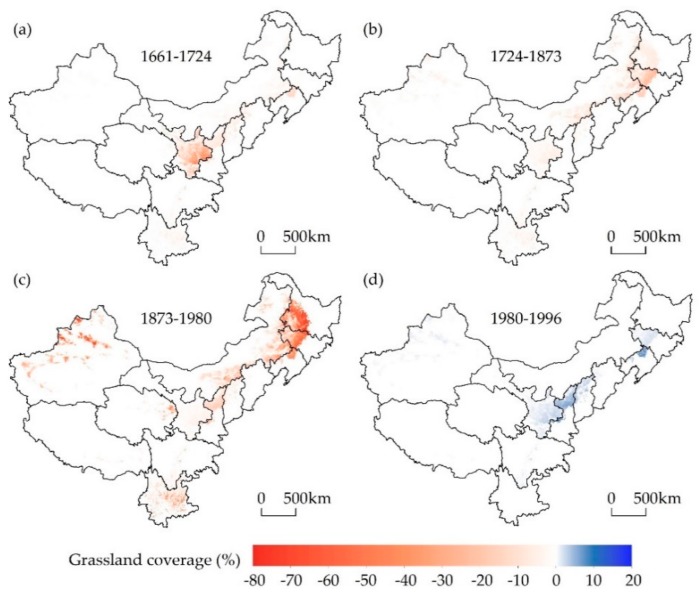
Maps showing net changes in grassland cover across western China for time periods between (**a**) 1661 and 1724; (**b**) 1724 and 1873; (**c**) 1873 and 1980; and (**d**) 1980 and 1996.

**Figure 8 ijerph-16-03160-f008:**
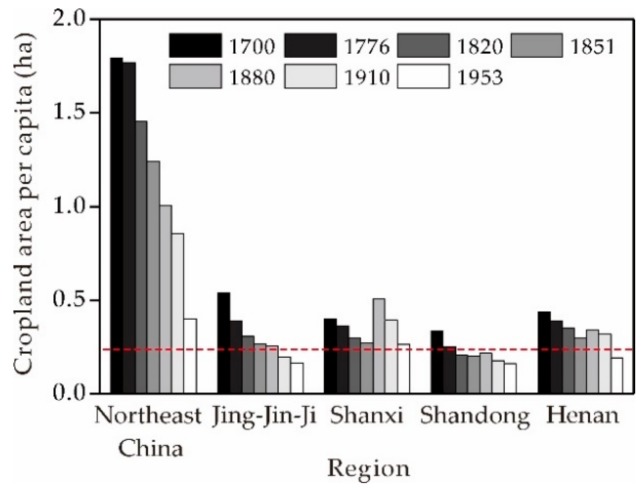
PCA values for different regions between 1700 and 1953. The red dashed line on this graph denotes minimum PCA, 0.27 ha [66,67]. Human population and cropland data were sourced from references [37,62,68].

**Figure 9 ijerph-16-03160-f009:**
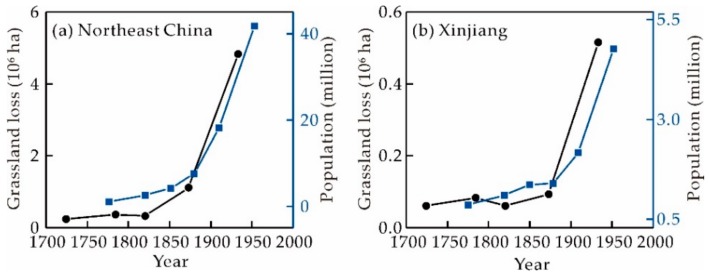
Population and grassland losses for the period between 1776 and 1953 reconstructed using PENG and CHCD for (**a**) northeastern China and (**b**) Xinjiang. Human population data were sourced from references [62,68].

**Figure 10 ijerph-16-03160-f010:**
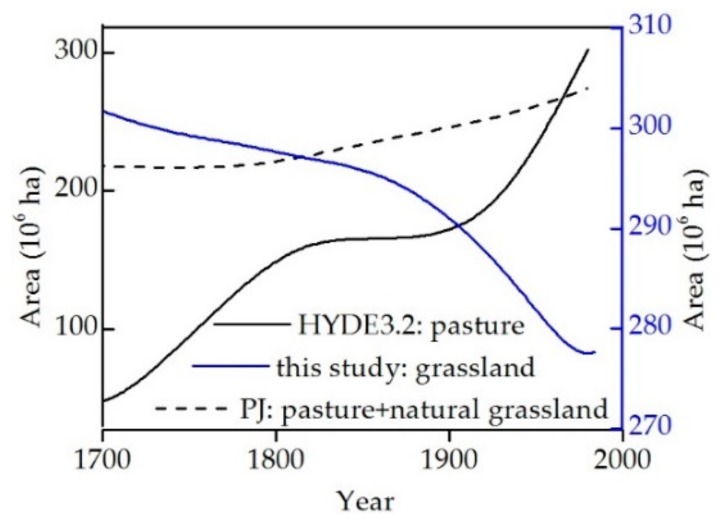
Trends in grassland and pasture area changes across western China over the last 300 years obtained from HYDE 3.2, PJ, and presented in this paper.

**Table 1 ijerph-16-03160-t001:** Comparison between potential vegetation and native vegetation in northeastern China.

	Forests (%)	Grasslands (%)	Water Body (%)	Data Source
Potential vegetation	57	38	5	Ramankutty and Foley [15]
Native vegetation	55.2	42.2	1.6	Zhang et al. [48]

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
