# Peer review of "Exploring Spatiotemporal Pattern of Grassland Cover in Western China from 1661 to 1996"

_ijerph, 2019, doi:10.3390/ijerph16173160_

Round 1

Reviewer 1 Report

This manuscript estimates historical grassland cover in western China using land cover data and potential negation maps. This was done primarily by assessing modern grassland cover and extrapolating backward. Later, CHCD and PDENG are integrated, but it is not very evident how or why (see comments below). The text does a great job of assigning historical explanations to the changes in grassland cover over the temporal span of the reconstructed maps.

This paper is well-written (minus some grammatical issues, some of which I point out), and the connection between PDENG, modern grass cover, and historical events is intriguing. The products used were not generated in this paper, but instead this ms uses already published data sources for a novel application. Much of the discussion is located in Results rather than Discussion. And the CHCD is poorly explained - to the extent that I cannot comment on the soundness of this methodology. But otherwise, this is a good paper that I would be happy to review following major revisions. Some specific points follow:

15 - You are missing an "l". Not LUCC but LULCC

19 - You did not use RS data. You used an RS-derived product. Be explicit.

20 - potential distribution extent of natural grassland vegetation (PDENG) > potential distribution of natural grassland vegetation (PDONG) or potential extent of natural grassland vegetation (PENG)

42-44 - awkward first sentence

43 - "lead to climate change" poor choice of words. please revise with more explicit wording

46 - The data products for this study do not appear to be at a high spatial resolution

48-50 - poor grammar

72-74 - unclear how the second sentence logically follows from the first. Consider clarifying.

75 - data > date(?)

81- delete "It is well known that"

83- delete "in the same way that is common internationally"

99 - "researches" is not a word in English

105-107 - passive tense. convert to active tense

122 - "apart from in" unclear and poor grammer

123 - why "(excluding southwest China)"? Northern Yunnan is sparely populated

139 - what is the spatial resolution of the CHCD?

145 - use of the word "respectively" does not make sense here.

149 - was some analysis conducted? what are the results that prove the data is reliable?

153 - what are the details on the CLUDs? What is their origin and spatial resolution?

160 - what was "stabilized"?

167-168 - This sentence does not follow logically from past ones. Please explain.

177-179 - what is the evidence for this? Does the reference support this claim?

179 - "dominated by land use" doesn't make sense. Do you mean anthropogenic land use?

181-183 - I think a lot more is needed to support the claim the modern grasslands were necessarily historically so (minus potential forest).

185-192 - this paragraph should be written more clearly

199-201 - this is a Discussion, not a Methods, point

204-206 - this is a Discussion, not a Methods, point. In any case, where are the results? Where is the evidence?

205 - 208 - this is unclear. I do not understand why this assumption is made. This section needs a significant revision to be explicit about what is being done and assumed, and why.

206-208 - This Boolean condition and the 90% threshold seems arbitrary. Please explain rationale.

228-229 - how were these dates determined?

232-234 - needs reference

237-239 - needs reference

271 - is there a smoothing function in these graphs?

277 - How are you getting grass cover %? Are these at 10km? If so, what account for this sib pixel %?

289-294 - needs reference

304 - "contradictions" is not the correct word. Is it "maodun" you are translating? If so, think of an alternate word in English.

323- reference?

348 - "means" is not correct. correlation does not equal causation

272-317 - this section belongs in Discussion

359-360 - There exist extremely large ag regions in both continents. Climate is very conducive to farming. Please qualify this statement with something that speaks of the specific small regions where farming has been replaced with animal husbandry.

463-482 - in general this conclusion reads like a recap of the results. A conclusion should tie results and discussion together in novel ways, with a clear path to broader implications

Author Response

Manuscript ID: ijerph-570233

Title: Exploring spatiotemporal pattern of grassland cover in Western China from 1661 to 1996 using remote sensing and cropland data

Response to Reviewer 1 Comments

Comments and Suggestions for Authors:

This manuscript estimates historical grassland cover in western China using land cover data and potential natural maps. This was done primarily by assessing modern grassland cover and extrapolating backward. Later, CHCD and PDENG are integrated, but it is not very evident how or why (see comments below). The text does a great job of assigning historical explanations to the changes in grassland cover over the temporal span of the reconstructed maps. This paper is well-written (minus some grammatical issues, some of which I point out), and the connection between PDENG, modern grass cover, and historical events is intriguing. The products used were not generated in this paper, but instead this ms uses already published data sources for a novel application. Much of the discussion is located in Results rather than Discussion. And the CHCD is poorly explained - to the extent that I cannot comment on the soundness of this methodology. But otherwise, this is a good paper that I would be happy to review following major revisions.

Response: Thank you for this comment. We revised or rewrote the Abstract (Line 14-35), Introduction (Line 40-106), Materials and Methods (Line 108-251), Discussion (Line 325-483) and Conclusions (Line 485-501). More literature review works were conducted and more references were cited to support our points.

In Materials and Methods section, we added more information to describe the details on CHCD, PNV, and CLUDs (Line 138-167). We tried our best to clarify our method in the section “2.3. Methodology” and the expression was rewritten (Line 169-251).

In Discussion section, we rewrote section 4.1 (Forces driving grassland changes) by combining section 3.3 (Grassland cover spatial patterns) with the original section 4.1 (Line 325-381). More references were cited to support our point of view.

(Note: the red font refers to that our expression has been modified in the revised manuscript, and the blue font represents the adjustment of its position.)

Some specific points follow:

Point 1: line 15 - You are missing an "l". Not LUCC but LULCC

Response 1: Following this comment, we revised the abbreviation “LUCC” to “LULCC” (Line 18). Meanwhile, we also checked the whole manuscript and revised this abbreviation in the same way.

Point 2: line 19 - You did not use RS data. You used an RS-derived product. Be explicit.

Response 2: Yes, agreed. Following this comment, we revise this expression to “remote sensing-derived LULCC data” (Line 17-18). And we perform a thorough revision of the manuscript to explicit this expression.

Point 3: line 20 - potential distribution extent of natural grassland vegetation (PDENG) > potential distribution of natural grassland vegetation (PDONG) or potential extent of natural grassland vegetation (PENG)

Response 3: Following this comment, we revised our expression “potential distribution extent of natural grassland vegetation (PDENG)” to “potential extent of natural grassland vegetation (PENG)”. And we perform a thorough revision of the manuscript to explicit this expression.

Point 4: line 42-44 - awkward first sentence. line 43 - "lead to climate change" poor choice of words. please revise with more explicit wording.

Response 4: Thank you for the suggestion. Following this comment, we modified this expression to “Human activities over long time periods have significantly altered the landscape of the Earth's surface by transforming natural ecosystems to agricultural areas [1,2]. These modifications have led to serious degradation of the ecological environment and have continuously influenced climate change and human sustainable development [3–6].” (Line 40-43).

Point 5: line 46 - The data products for this study do not appear to be at a high spatial resolution

Response 5: Following this comment, we deleted the phrase “high spatial resolution” (Line 45).

Point 6: line 48-50 - poor grammar

Response 6: Following this comment, we rewrote our sentence “Indeed, historical LULCC studies have gradually moved past their previous limitations (i.e., the results have been qualitative, statistically insignificant); spatial explicit reconstruction results have been widely utilized in modeling climate change, carbon emissions, and the ecological environmental effects due to human activities [7–10].” (Line 46-50).

Point 7: line 72-74 - unclear how the second sentence logically follows from the first. Consider clarifying.

Response 7: As you commented, it is not well connected between the preceding and proceeding sentences, so we revised our expression. Specifically, we removed this vague description and added more information (Line 81-82, 88-89).

Point 8: line 75 - data > date(?)

Response 8: As you commented, we revised the word “data” to “date”. And we perform a thorough revision of the manuscript to explicit this expression.

Point 9: line 81- delete "It is well known that"

Response 9: Thank you. Revised. (Line 70).

Point 10: line 83- delete "in the same way that is common internationally"

Response 10: Thank you for the suggestion. Revised. (Line 71).

Point 11: line 99 - "researches" is not a word in English

Response 11: Following this comment, we revised the word “researches” to “progresses” (Line 96).

Point 12: line 105-107 - passive tense. convert to active tense

Response 12: Thank you for the suggestion. We rewrote our sentence “Specifically, building on an understanding of the characteristics and driving forces of grassland change in western China, we propose a method for reconstructing historical grassland cover across this region and further map grassland cover between 1661 and 1996.” (Line 104-106).

Point 13: line 122 - "apart from in" unclear and poor grammer

Response 13: Following this comment, we deleted the phrase “apart from” and modified our sentence “Average annual precipitation across the whole region is less than 600 mm (excluding Chuan-Yu and Yunnan).” (Line 120-121).

Point 14: line 123 - why "(excluding southwest China)"? Northern Yunnan is sparely populated

Response 14: The expression of this sentence is ambiguous. Following this comment, we revised the sentence “Historically, compared with eastern China, this study area is sparsely populated and has been rich in grassland resources.” (Line 121-123).

Point 15: line 139 - what is the spatial resolution of the CHCD?

Response 15: The spatial resolution of the CHCD is 10km. Following this comment, we clarified a bit further (Line 139).

Point 16: line 145 - use of the word "respectively" does not make sense here.

Response 16: Thank you. Following this comment, we deleted the word “respectively” (Line 146-147).

Point 17: line 149 - was some analysis conducted? what are the results that prove the data is reliable?

Response 17: Following this comment, we tried our best to prove that the data is reliable. Specifically, we collected more information and data on potential vegetation and native vegetation and the correlation between them were presented in terms of spatial pattern and grassland coverage. And we added a table (Table 1) and a corresponding description in the revised manuscript to illustrate this question (Line 148-155).

Point 18: line 153 - what are the details on the CLUDs? What is their origin and spatial resolution?

Response 18: Following this comment, we provided more information about the land use/cover datasets, especially their origin and spatial resolution (Line 156-167). These datasets are publicly released data products that can be used directly.

Specifically, a series of RS-derived land use/cover data of western China encompassing the period between 1980 and 2015 were obtained from Chinese LULCC datasets (CLUDs, available from http://www.resdc.cn). Liu et al. [1,2] use human-computer interactive interpretation method as well as Landsat TM/ETM digital images covering China to construct CLUDs. The land use/cover types in CLUDs include six classes such as cropland, forestland, grassland, water body, unused land, and built-up land, and 25 subclasses. These CLUDs, with a resolution of 1 km, were updated regularly at 5-year intervals from 1980 to 2015 and the quality of datasets was ensured through uniform quality control and integration checking. In order to assess the accuracy of the database, a large number of field-investigation records were obtained. Results indicate the accuracy of the six classes of land use/cover type was greater than 94.3%, and the overall accuracy of the 25 subclasses was greater than 91.2%, which can satisfy the demands of user mapping accuracy at a scale of 1:1 million.

References

Liu, J.Y.; Kuang, W.H.; Zhang, Z.X.; Xu, X.L.; Qin, Y.W.; Ning, J.; Zhou, W.C.; Zhang, S.W.; Dong, L.R.; Zhen, Y.C., et al. Spatiotemporal characteristics, patterns, and causes of land-use changes in China since the late 1980. J. Geogr. Sci. 2014, 24, 195–210. Ning, J.; Liu, J.Y.; Kuang, W.H.; Xu, X.L.; Zhang, S.W.; Yan, C.Z.; Li, R.D.; Wu, S.X.; Hu, Y.F.; Du, G.M., et al. Spatio-temporal patterns and characteristics of land-use change in China during 2010–2015. J. Geogr. Sci. 2018, 28, 547–562.

Point 19: line 160 - what was "stabilized"?

Response 19: Following this comment, we removed this vague description “stabilized” and rewrote our sentence “In the course of safeguarding the security of borderlands in western China, grassland was often regarded as an important agricultural reserve resource by successive dynastic governments.” (Line 173-174).

Point 20: line 167-168 - This sentence does not follow logically from past ones. Please explain.

Response 20: Following this comment, we rewrote our sentence “This means that available and reliable cropland data across this region can be used to reflect reclamation of grassland because these data are the quantitative expression of land reclamation behavior.” (Line178-182).

Specifically, Line 168-182: we elaborated that land reclamation was the main driving force of grassland cover change in western China historically. We planned to use land reclamation to extrapolate backward grassland cover change, given that historical cropland data are available and reliable. Indeed, land reclamation is a kind of anthropogenic land use behaviors; cropland data are the quantitative expression of this behavior.

Point 21: 177-179 - what is the evidence for this? Does the reference support this claim?

Response 21: Following this comment, we rewrote section 2.3.1 “The PENG in the absence of human land use” and added more references and information (Line190-219).

For the application using land reclamation to extrapolate backward grassland cover change, it is essential to know the land surface properties of western China before agriculture emerged. The native vegetation reconstructed based on historical evidence (e.g., historical archives, pollen, or archaeological observations) can objectively reflect the land cover before land reclamation [1]. But the reconstruction results only provide a macro-spatial pattern [2], which cannot meet the demands of exploring land cover change on a large spatial scale. In this case, scholars generally replace native vegetation with potential vegetation as a starting point for analyzing land cover change [3,4].

The main reconstruction methods on PNV are as follows: 1) Extrapolation of existing natural vegetation to similar habitats (a simple method of qualitative estimation) [5]; 2) Simulating PNV by using the relationship between vegetation and environment [6,7]; 3) synthesizing 1) and 2) to determine PNV. The third method is currently widely used in the historical LUCC reconstruction field [3,4,8,9]. Therefore, we adopt this method to determine PNV and make necessary adjustments and supplements according to the characteristics of western China. Indeed, Lan [10] pointed out that small parts of the modern grassland area might also be forested in the upper Yangtze River historically, and the current grassland vegetation is the result of the interaction of human activities and natural environment. It is appropriate to assume that most modern grassland regions detected using RS across western China were also grassland regions historically and, indeed, small parts might also be forested; modern non-grassland regions monitored by RS (i.e. those dominated by land use, including cultivated regions and built-up areas) might also have been grasslands in the past.

Specifically, we extracted the distributions of modern natural vegetation (forest, grassland) and anthropogenic land cover (cultivated regions and built-up areas) in western China using CLUDs, while potential grassland and forest areas were obtained from the PNV data by Ramankutty and Foley [3]. After grid cells of potential forest were subtracted from modern grassland cover, the remainder were classified as both the past and present grassland distributional area. Superimposing potential grassland cover over that seen currently non-grassland cover, the overlapped grid cells enabled the determination of historical natural grassland cover in the non-grassland regions. Finally, combining these maps for modern grassland and the non-grassland regions enabled the determination of the PENG in the absence of land reclamation.

References

Zhang, X.Z.; Wang, W.Q.; Fang, X.Q. Natural vegetation pattern over northeast China in late 17th century. Sci. Geogr. Sin. 2011, 31, 184–189. Wang, W.M.; Li, C.H.; Shu, J.W.; Chen, W. Changes of vegetation in southern China. Science China Earth Sciences 2019., 62, 1316-1328, doi:10.1007/s11430-018-9364-9. Ramankutty, N.; Foley, J.A. Estimating historical changes in global land cover: Croplands from 1700 to 1992. Glob. Biogeochem. Cycle 1999, 13, 997–1027. Pongratz, J.; Reick, C.; Raddatz, T.; Claussen, M. A reconstruction of global agricultural areas and land cover for the last millennium. Glob. Biogeochem. Cycle 2008, 22, GB3018. Stumpel, A.H.P.; Kalkhoven, J.T.R.; Leersum; Stumpel-Rienks, S.E.; van der Maarel, E. A vegetation map of The Netherlands, based on the relationship between ecotopes and types of potential natural vegetation. Vegetatio 1978, 37, 163-173, doi:10.1007/bf00717650. Haxeltine, A.; Prentice, I.C. BIOME3: An equilibrium terrestrial biosphere model based on ecophysiological constraints, resource availability, and competition among plant functional types. Global Biogeochemical Cycles 1996, 10, 693-709, doi:10.1029/96gb02344. Ni, J.; Sykes, M.T.; Prentice, I.C.; Cramer, W. Modelling the vegetation of china using the process-based equilibrium terrestrial biosphere model biome3. Glob Ecol Biogeogr 2000, 9, 463-479. He, F.N.; Li, S.C.; Zhang, X.Z. A spatially explicit reconstruction of forest cover in China over 1700–2000. Glob. Planet. Change 2015, 131, 73–81. Yang, X.H.; Jin, X.B.; Xiang, X.M.; Fan, Y.T.; Shan, W.; Zhou, Y.K. Reconstructing the spatial pattern of historical forest land in China in the past 300 years. Glob. Planet. Change 2018, 165, 173–185. Lan, Y. Vegetation succession of the medium and low mountainous areas along the subtropics of the upper Yangtze River in recent 500 years. Geogr Res. 2010, 29, 1182–1192.

Point 22: line 179 - "dominated by land use" doesn't make sense. Do you mean anthropogenic land use?

Response 22: Thank you for the suggestion. Revised. (Line 209).

Point 23: line 181-183 - I think a lot more is needed to support the claim the modern grasslands were necessarily historically so (minus potential forest).

Response 23: Following this comment, we tried our best to support the claim. Specifically, we added more descriptions and references to clarify the origin of the method, the rationality of the hypothesis, and the improvements we have made for the western region (Section 2.3.1, Line190-219).

Point 24: line 185-192 - this paragraph should be written more clearly

Response 24: Following this comment, we revise this expression (Line 211-219).

Specifically, we extracted the distributions of modern natural vegetation (forest, grassland) and anthropogenic land cover (cultivated regions and built-up areas) in western China from CLUDs, while potential grassland and forest areas were obtained from the PNV data by Ramankutty and Foley [15]. After grid cells of potential forest were subtracted from modern grassland cover, the remainder were classified as both the past and present grassland distributional area. Superimposing potential grassland cover over that seen currently non-grassland cover, the overlapped grid cells were classified as historical natural grassland cover in the non-grassland regions. Finally, combining these maps for modern grassland and the non-grassland regions enabled the determination of the PENG in the absence of land reclamation.

Point 25: line 199-201 - this is a Discussion, not a Methods, point; line 204-206 - this is a Discussion, not a Methods, point. In any case, where are the results? Where is the evidence?

Response 25: Following this comment, we rewrote section “2.3.2. Reconstructing grassland cover”. And Figure 3 was illustrated to show the statistical relationship between cropland cover and grassland cover at grid scale. We clarified the correlation between cropland and grassland both in the past and at present (Line 220-230).

Specifically, population growth has long been considered as the main driver of LULCC [21,54]. As population increases in a given region, so does demand for food. Croplands are therefore expanded to provide more food. Because of the backward productive level in historical times, the increase of grain yields was mainly due to the expansion of cropland [19,21]. For western China, the increase of cropland has encroached on a large amount of grassland cover. And the cropland in this region is mainly distributed around urban built-up areas and rural settlements both in the past and at present. As the amount of cropland increases along with population growth, residential land area also expands. Statistical results of CLUDs show that grassland coverage decreased as cultivated land area increased; as the proportion of cropland was greater than, or equal to, 90%, corresponding grassland coverage was mostly less than 1% because the remainder was usually occupied by residential land (Figure 3).

Point 26: line 205 - 208 - this is unclear. I do not understand why this assumption is made. This section needs a significant revision to be explicit about what is being done and assumed, and why.

Response 26: Following this comment, we rewrote section 2.3.2. In the first paragraph of section 2.3.2 (line 221-230), we clarified the correlation between cropland and grassland both in the past and at present. Then, we presented the supporting information, e.g., Figure 3, for the hypothesis (Line 233-240).

Specifically, modern LULCC information can be obtained through monitoring of satellite remote sensing, aerial photographs, and field surveys; past LULCC is, however, limited in space and time due to the lack of direct, large-scale observations [57]. Therefore, the common method used by historical LULCC reconstruction scholars is to set some reasonable assumptions and then propose a past LULCC scenario based on the relevant knowledge of modern LULCC [15,18-21]. Based on the above analysis, we believe that the correlation between modern cropland and grassland shown in Figure 3 is also applicable to the past. This study therefore assumed that historical grassland coverage was zero in a grid if the proportion of cropland was greater than 90% in western China.

Point 27: line 206-208 - This Boolean condition and the 90% threshold seems arbitrary. Please explain rationale.

Response 27: Following this comment, we rewrote sections “2.3.1. The PENG in the absence of human land use” and “2.3.2. Reconstructing grassland cover”. Among them, this Boolean condition is a specific embodiment of the potential vegetation reconstruction methods (See line 220-230 for details). The 90% threshold was explicitly expressed through the addition of Figure 3 (See line 231-232 for details).

Point 28: line 228-229 - how were these dates determined?

Response 28: In this paper, we extracted cropland data from CHCD across this region to reconstruct grassland cover. Because the CHCD is available for the years 1661, 1724, 1784, 1820, 1873, 1933, 1980, and 1996, so grassland data have the same dates as CHCD. Then, on the basis of the changing trend of total grassland area, we divided the whole study period into four phases.

Point 29: line 232-234 - needs reference

Response 29: Following this comment, we added relevant references to support our point of view (Line 266).

References

Miao, Y.; Lu, X.S. Primary study on the reasons of grassland reclamation of China during the historical period. Pratacult Sci. 2008 25, 124-129. Meng, F.G. Agricultural Development in Gansu province in the early Qing Dynasty and Its historical reflection. J. Lanzhou Acad. 2009, 74-77.

Point 30: line 237-239 - needs reference

Response 30: Thank you. Following this comment, we added relevant references to support our point of view (Line 271-272).

References

Wang, X.L. Northwest Mita in Qing Dynasty; Xinjiang People's Press: Urumqi, China, 2012. Deng W.Y. Study on stationing troops to open up wasteland in Western Regions and development of Xinjiang; Guangdong People's Press: Guangzhou, China, 2017. Cao, S.J. Population History of China. Vol. 5, Qing Dynasty Period; Fudan University Press: Shanghai, China, 2001.

Point 31: line 271 - is there a smoothing function in these graphs?

Response 31: Yes. Data curves were smoothed using B-spline function in origin 9.0 software. Following this comment, we added the relevant information “Data curves were smoothed using B-spline function.” (Line 303-304).

Point 32: line 277 - How are you getting grass cover %? Are these at 10km? If so, what account for this sib pixel %?

Response 32: The spatial resolution of grassland cover is 10km × 10km, so the area of a grid is 100 km2. Therefore, this grassland coverage was expressed as follows at the grid scale:

In this expression, i refers to a grid (10 km) in year t. So this pixel value refers to the proportion of grassland area in the grid (10 km). This expression (%) was also used in the following references.

References

Yang, X.H.; Jin, X.B.; Xiang, X.M.; Fan, Y.T.; Shan, W.; Zhou, Y.K. Reconstructing the spatial pattern of historical forest land in China in the past 300 years. Glob. Planet. Change 2018, 165, 173–185. Wei, X.; Ye, Y.; Zhang, Q.; Li, B.; Wei, Z. Reconstruction of cropland change in North China Plain Area over the past 300 years. Global and Planetary Change 2019, 176, 60-70.

Point 33: line 289-294 - needs reference

Response 34: Revised. This part has been moved to section “Discussion” (Line 331).

Point 34: line 304 - "contradictions" is not the correct word. Is it "maodun" you are translating? If so, think of an alternate word in English.

Response 34: Thank you for the suggestion. We revised the word “contradictions” to “conflicts” (Line 343).

“Throughout the late Qing Dynasty period, conflicts between rapid population increases in the traditionally cultivated regions of China and limitations in such land resources coupled with natural disasters and chaos caused by wars became major societal problems.”

Point 35: line 323- reference?

Response 35: Thank you. Revised. (Line 347).

Point 36: line 348 - "means" is not correct. correlation does not equal causation

Response 36: Yes, agreed. Following this comment, we deleted the vague expression and rewrote section “4.1. Forces driving grassland changes” (Line 325-381).

Point 37: line 272-317 - this section belongs in Discussion

Response 37: Yes, agreed. Thank you for the suggestion. Following this comment, we rewrote section 4.1 by combining section 3.3 with the original section 4.1 (Line 325-381).

Point 38: line 359-360 - There exist extremely large ag regions in both continents. Climate is very conducive to farming. Please qualify this statement with something that speaks of the specific small regions where farming has been replaced with animal husbandry.

Response 38: Following this comment, we added new references and modified our expression to support this point (Line 389-393).

Specifically, For Europe and the United States, there has been such a high proportion of animal husbandry and a large amount of pasture over a long period of time especially in the Alps of central Europe and the western mountain areas of the United States [1]. Land use strategy in these regions is that population growth stimulates the development of animal husbandry and therefore enlarges pasture areas [2,3].

References

Ramankutty, N.; Evan, A.T.; Monfreda, C.; Foley, J.A. Farming the planet: 1. Geographic distribution of global agricultural lands in the year 2000. Global Biogeochemical Cycles 2008, 22, GB1003, doi:10.1029/2007gb002952. 2. Morrison, K.D. Provincializing the Anthropocene. Seminar 2013, 676, 75–80. He, F.N.; Li, S.C.; Yang, F.; Li, M.J. Evaluating the accuracy of Chinese pasture data in global historical land use datasets. Sci. China-Earth Sci. 2018, 61, 1685–1696.

Point 39: line 463-482 - in general this conclusion reads like a recap of the results. A conclusion should tie results and discussion together in novel ways, with a clear path to broader implications

Response 39: Following this comment, we rewrote section “5. Conclusions” by tying the results and discussion (Line 485-501).

Specifically, based on grassland definition and land use practice in western China, this study presents a reconstruction-based model for historical grassland cover using information from PENG and CHCD. Grassland cover maps for western China were generated at a 10 km resolution for the period between 1661 and 1996. Results suggest that the natural grassland area across western China (without land reclamation) was originally 320.72 × 106 ha; models imply that decreasing trends were also observed in all western provinces from 1661 to 1996.

Grassland cultivation mainly occurred within the Gan-Ning area and was driven by agricultural development policies of the Qing government between 1661 and 1724. Agriculture was subsequently expanded in Inner Mongolia and northeastern China due to an increased number of refugees; this led to a slight decline in grassland area between 1724 and 1873. Over the period between 1873 and 1980, opening up of grasslands was mainly concentrated in northeastern China and Xinjiang. Large increases in both the intensity and extent of reclaimed grassland resulted from agricultural economic crisis, policies, and exponentially increased immigration.

Definitions for pasture, land use practice, and the reconstruction methods applied by global datasets like HYDE 3.2 and PJ cannot be considered suitable for western China. Data from previous compilations fail to objectively reveal the influence of long-term anthropogenic activities on grassland vegetation of western China.

Reviewer 2 Report

Dear authors, this paper has collected a lot of pertinent data, and deals with a large area and a ver long time period, and therefore has a lot of potential to be relevant contribution to science. However, to be publishable, it needs a lot more work, mostly in terms of structure and order of information. Also, it is not clear what the actual aims and objectives are. Is the main aim of the paper to assess the level of correctness of global historical grassland cover datasets to the situation in China, or is their aim to study the LULC, if it’s the former it needs restructuring the paper, if it’s the latter, it needs different, more precise and streamlined methods. Is the main goal to analyse the contributing factors of land cover change?

Please find more detailed comments in the text:

Abstract

Introduction in the abstract a bit wide and unnecessary (14-18) and could be rewritten. More details is needed on the data – “a series of remotely sensed data” is not precise enough, nor is  alongside “potential natural vegetation data”. At least the source of this data needs to be briefly mentioned in the abstract.

Lines 49-50: results obtained to data (to date?) have been qualitative, statistically

50 “insignificant, and therefore hard to utilize in follow-up simulations on environmental change”. You need to support this with references.

Line 73 inconsistencies in definitions of what

Line 81 – “it is well known” is not necessary delete it

83-92 talks about the importance of grasslands, after the global datasets on grasslands have been introduced. The importance of grasslands in china is a more general topic and needs to be moved to the earlier parts of the introduction. Introduction needs to go from a very general perspective to a narrower perspective, and needs to finish with a clear research question, aim and objectives. Aims in this paper are too vague (to reconstruct historical grassland cover across western China – for which period)? There are no objectives.

Line 110 data in study area without any references – where are we getting data on grassland from?

152-154 not enough details provided. Methods need to be reproducible.

Is the entire research here from secondary sources? A compilation of existing LULCC datasets? If yes, needs to be mentioned sooner rather than later.

Very confusing methods section.

Order of information is not adequate – what goes to what part – 156 do 168, needs to be put in “study area”

Figure 2 is a good figure

2.3.2. presenting some results in the methods section, why? – 199 to 201

Line 232 233 already discussion, but no references 244-246 289-292 295 312

Most of the historical data here are also supported with very few references. There is no comparison with relevant studies.

359-366 not clear how relevant necessary just two references for the first part of the paragraph, none for the second.

Discussion:

The discussion chapter needs to use much more relevant literature in trying to explain the findings stemming from this research.

Entering a new variable – PCA – doesn’t make sense in the discussion. It probably should have been added to the methods and results as a potential indicator of drivers of grassland change, and then used in the discussion. Was the aim to map the historical grassland use change, or was the aim to assess the potential contributing factors of change as well? If it’s the latter, the changes need to be implemented throughout.

Furthermore, you calculated correlation between population loss and grassland cover change, but it was not mentioned in the methods or the results sections, so again if it’s an attempt to analyse the contributing factors, the changes in the text need to be implemented throughout. An important note here – correlation does not imply causation, and the established correlation needs to be explained by using relevant literature. Your established correlations just should just suggest possible causation.

359-361 the mentioned references do not support what was said in the text, and the remainder of the paragraph is not supported with a single reference. The use of existing literature needs to be much clearer and consistent, but also pertinent to the topic.

Line 382-386 discuss the definition of grasslands in this research. This would probably be better in introduction or methods section.

Chapter 4.2. needs to present the main findings in a clear manner. What is the main reason for the stark difference in the results presented in Figure 9?

Generally, 4.2., 4.3. and 4.4. are an interesting contribution, but it was not one of the aims or objectives of the paper, so it is not clear how it fits in the paper overall. You need to be clear at what you are trying to present and why. Is the aim to find trends in grassland change? Or is it to assess the drivers of change? Or is it to compare the quality of different global datasets and assess it to local/regional level?

Conclusion looks like results – too specific repetition of the actual results. Most important findings need to be reiterated generally and put into wider context.

Author Response

Manuscript ID: ijerph-570233

Title: Exploring spatiotemporal pattern of grassland cover in Western China from 1661 to 1996 using remote sensing and cropland data

Response to Reviewer 2 Comments

Comments and Suggestions for Authors

Dear authors, this paper has collected a lot of pertinent data, and deals with a large area and a ver long time period, and therefore has a lot of potential to be relevant contribution to science. However, to be publishable, it needs a lot more work, mostly in terms of structure and order of information.

Also, it is not clear what the actual aims and objectives are. Is the main aim of the paper to assess the level of correctness of global historical grassland cover datasets to the situation in China, or is their aim to study the LULC, if it’s the former it needs restructuring the paper, if it’s the latter, it needs different, more precise and streamlined methods. Is the main goal to analyse the contributing factors of land cover change?

Response: Thank you for this comment. This study aims to reconstruct historical grassland cover across western China over the past 300 years. We rewrote the Abstract (Line 14-35), Introduction (Line 40-106), Materials and Methods (Line 107-251), Discussion (Line 323-483) and Conclusions (Line 485-501) to make our main aim clearer. More literature review works were conducted and more references were cited to support our points.

In Introduction section, according to the principle from a very general perspective to a narrower perspective, we rewrote section “Introduction” (Line 40-106). And this study aims to reconstruct historical grassland cover across western China over the past 300 years (Line 102-103).

In Materials and Methods section, we added more information to describe the details on CHCD, PNV, and CLUDs (Line 137-167). We tried our best to clarify our method in section “2.3. Methodology” and the expression was rewritten (Line 168-251).

In Discussion section, we rewrote section 4.1 (Forces driving grassland changes) by combining section 3.3 (Grassland cover spatial patterns) with the original section 4.1 (Line 325-381). More references were cited to support our point of view.

(Note: the red font refers to that our expression has been modified in the revised manuscript, and the blue font represents the adjustment of its position.)

Please find more detailed comments in the text:

Abstract

Point 1: Introduction in the abstract a bit wide and unnecessary (14-18) and could be rewritten.

Response 1: Thank you for the suggestion. Following this comment, we rewrote our expression “Historical grassland cover change is an example of primary historical land use and land cover change (LULCC) and can provide significant basic data for global and regional environmental change modeling.” (Line 14-15).

Point 2: More details is needed on the data – “a series of remotely sensed data” is not precise enough, nor is alongside “potential natural vegetation data”. At least the source of this data needs to be briefly mentioned in the abstract.

Response 2: Yes, agreed. Following this comment, we modified our expression “This study presents a method for reconstructing grassland cover over the past 300 years. By synthesizing remote sensing-derived Chinese land use and land cover change (LULCC) data (1980-2015) and potential natural vegetation data simulated by the relationship between vegetation and environment, we first determined the potential extent of natural grassland vegetation (PENG) in the absence of human activities. Then we reconstructed grassland cover across western China between 1661 and 1996 at 10 km resolution by overlaying the Chinese historical cropland dataset (CHCD) over the PENG. (Line 16-22).

Point 3: Lines 49-50: results obtained to data (to date?) have been qualitative, statistically

50 “insignificant, and therefore hard to utilize in follow-up simulations on environmental change”. You need to support this with references.

Response 3: Thank you. As you commented, we revised the word “data” to “date”. We rewrote the sentence and added more references to support this point (Line 46-50).

Specifically, Indeed, historical LULCC studies have gradually moved past their previous limitations (i.e., the results to date have been qualitative, statistically insignificant); newly spatial explicit reconstruction results have been widely utilized in modeling climate change, carbon emissions, and the ecological environmental effects due to human activities [7–10].

Point 4: Line 73 inconsistencies in definitions of what

Response 4: Following this comment, we revised our expression in this paragraph and adjusted its position. Specifically, we removed this vague description and added more information (Line 81-82, 88-89).

Point 5: Line 81 – “it is well known” is not necessary delete it

Response 5: Thank you for the suggestion. Revised.

Point 6: 83-92 talks about the importance of grasslands, after the global datasets on grasslands have been introduced. The importance of grasslands in china is a more general topic and needs to be moved to the earlier parts of the introduction. Introduction needs to go from a very general perspective to a narrower perspective, and needs to finish with a clear research question, aim and objectives. Aims in this paper are too vague (to reconstruct historical grassland cover across western China – for which period)? There are no objectives.

Response 6: Thank you for the suggestion. Following this comment, according to the principle from a very general perspective to a narrower perspective, we rewrote section “Introduction”. The red font refers to that our expression has been modified, and the blue font represents the adjustment of its position.

Specifically,

Paragraph 1. The importance of historical land use and land cover change (LULCC); (Line 40-52). Paragraph 2. The importance of grasslands and the global datasets on grasslands; (Line 53-69). Paragraph 3. Grasslands in China; (Line 70-80). Paragraph 4. The uncertainties of global datasets including Chinese historical grassland data; (Line 81-89). Paragraph 5. The gaps of grassland reconstruction in China; (Line 90-99). Paragraph 6. The objective of this paper (Line 100-106). Moreover, this paragraph was rewritten to clarify the objective.

Point 7: Line 110 data in study area without any references – where are we getting data on grassland from?

Response 7: Thank you. Revised. (Line 109).

References

Su, D.X. The regional distribution and productivity structure of the Chinese grassland resources. Acta Agrestia Sinica 1994, 2, 71-77.

Point 8: 152-154 not enough details provided. Methods need to be reproducible. Is the entire research here from secondary sources? A compilation of existing LULCC datasets? If yes, needs to be mentioned sooner rather than later.

Response 8: Following this comment, we rewrote section “2.2. Data sources” and added more information on data sources (Line 137-167).

Three datasets were given in this section.

1) The Chinese historical cropland dataset (CHCD) (1661-1996); (Line 137-147).

2) The potential natural vegetation data (PNV); (Line 148-155).

3) Chinese land use /cover datasets (CLUDs) (1980-2015). (Line 156-167).

All of these datasets play an important role in reconstructing historical grassland cover. Specifically, we used the PNV and CLUDs to determine the potential extent of natural grassland vegetation (PENG) in the absence of human activities. Then, we derived grassland cover across western China between 1661 and 1996 at 10 km resolution by synthesizing the CHCD and the PENG.

Point 9: Very confusing methods section. Order of information is not adequate – what goes to what part – 156 do 168, needs to be put in “study area”

Response 9: Following this comment, we modified our expression (Line 173-174, 178-182).

The purpose of this part (Line 168-182) is to clarify land use strategy in western China historically, that is, utilization patterns of grassland resources. Specifically, as follows:

1) Over the last 2,000 years, western China has experienced four stages of large-scale immigration;

2) Grassland was often regarded as an important agricultural reserve resource by successive dynastic governments;

3) As the agricultural population has migrated from the mid-east into western China, agriculture has intensified and the range of land used for this function has expanded, resulting in the large-scale conversion of grasslands to cultivated land;

4) Changes in grassland cover have been significantly influenced historically by land reclamation.

Based on this knowledge, land use strategy means that more grassland is likely to be converted into cropland as population grows. So, we believe that available and reliable cropland data across this region can be used to reflect reclamation of grassland.

This part, an important part of methodology, clarified the theoretical basis of the reconstruction method.

Figure 2 is a good figure

Thank you for this comment.

Point 10: 2.3.2. presenting some results in the methods section, why? – 199 to 201

Response 10: Following this comment, we deleted the sentence and rewrote section “2.3.2. Reconstructing grassland cover” (Line 220-240).

Point 11: Line 232-233 already discussion, but no references 244-246 289-292 295 312

Most of the historical data here are also supported with very few references. There is no comparison with relevant studies.

Response 11: Following this comment, we added relevant references to support our point of view.

1) Line 232-233

References

Miao, Y.; Lu, X.S. Primary study on the reasons of grassland reclamation of China during the historical period. Pratacult Sci. 2008 25, 124-129. Meng, F.G. Agricultural Development in Gansu province in the early Qing Dynasty and Its historical reflection. J. Lanzhou Acad. 2009, 74-77.

2) Line 244-246

References

Zhang, X.L. The Impact of the Settlement Layout of the Qing Dynasty on the Security of the Northwestern Frontier. J. Shihezi Univ. (Philo and Soc. Sci) 2011, 25, 14–18.

3) Line 289-292

References

Miao, Y.; Lu, X.S. Primary study on the reasons of grassland reclamation of China during the historical period. Pratacult Sci. 2008 25, 124-129. Meng, F.G. Agricultural Development in Gansu province in the early Qing Dynasty and Its historical reflection. J. Lanzhou Acad. 2009, 74-77.

This part has been moved to section “Discussion” (Line 320).

4) Line 295

References

Li, W.; Zhang, P.Y.; Song, Y.X. Analysis on land development and causes in Northeast China during Qing dynasty. Sci. Geogr. Sin. 2005, 25, 7–16.

5) Line 312

References

Zhang, X.L. The Impact of the Settlement Layout of the Qing Dynasty on the Security of the Northwestern Frontier. J. Shihezi Univ. (Philo and Soc. Sci) 2011, 25, 14–18.

Point 12: 359-366 not clear how relevant necessary just two references for the first part of the paragraph, none for the second.

Response 12: Following this comment, we rewrote this paragraph and added new references to support the point (Line 389-393).

Specifically, for Europe and the United States, there has been such a high proportion of animal husbandry and a large amount of pasture over a long period of time especially in the Alps of central Europe and the western mountain areas of the United States [1]. Land use strategy in these regions is that population growth stimulates the development of animal husbandry and therefore enlarges pasture areas [2-4].

References

Ramankutty, N.; Evan, A.T.; Monfreda, C.; Foley, J.A. Farming the planet: 1. Geographic distribution of global agricultural lands in the year 2000. Global Biogeochemical Cycles 2008, 22, GB1003, doi:10.1029/2007gb002952. Klein Goldewijk, K.; Beusen, A.; Doelman, J.; Stehfest, E. Anthropogenic land use estimates for the Holocene - HYDE 3.2. Earth Syst. Sci. Data 2017, 9, 927–953. Pongratz, J.; Reick, C.; Raddatz, T.; Claussen, M. A reconstruction of global agricultural areas and land cover for the last millennium. Glob. Biogeochem. Cycle 2008, 22, GB3018. Kaplan, J.O.; Ruddiman, W.F.; Crucifix, M.C.; Oldfield, F.A.; Krumhardt, K.M.; Ellis, E.C.; Ruddiman, W.F.; Lemmen, C.; Goldewijk, K.K. Holocene carbon emissions as a result of anthropogenic land cover change. The Holocene 2011, 21, 775–791.

Discussion:

The discussion chapter needs to use much more relevant literature in trying to explain the findings stemming from this research.

Point 13: Entering a new variable – PCA – doesn’t make sense in the discussion. It probably should have been added to the methods and results as a potential indicator of drivers of grassland change, and then used in the discussion. Was the aim to map the historical grassland use change, or was the aim to assess the potential contributing factors of change as well? If it’s the latter, the changes need to be implemented throughout.

Response 13: Following this comment, we rewrote section 4.1 by combining section 3.3 with the original section 4.1 (Line 325-381).

This study aims to reconstruct historical grassland cover change. To evaluate whether the reconstruction results are reliable, section 4.1 analyzes the relationship between grassland resource dynamics and human factors such as population, economy, and policy and further assesses whether the results in this study coincide with sporadic historical data. Per capita cropland area (PCA) is often used to weigh food security in a country or region [64]. For the ancient agricultural society, the degree of food supply to the existing population is a significant indicator to measure the level of economic development of a country or region. Therefore, we used PCA to reflect the socioeconomic situation at that time.

Point 14: Furthermore, you calculated correlation between population loss and grassland cover change, but it was not mentioned in the methods or the results sections, so again if it’s an attempt to analyse the contributing factors, the changes in the text need to be implemented throughout. An important note here – correlation does not imply causation, and the established correlation needs to be explained by using relevant literature. Your established correlations just should just suggest possible causation.

Response 14: Following this comment, we rewrote section 4.1 (Line 325-381). As you commented, the correlation does not equal causation. Therefore, we modified it as follows:

1) (Economy) we used PCA to reflect the socioeconomic background. Results show population increase in this traditional agricultural region and the resultant agricultural economic crisis were therefore the main causes of a large number of refugees;

2) (Policy) the Qing government was forced to abolish its ‘Prohibit reclamation in Northeast China’ policy and encourage immigration into borderlands;

3) (Population) losses in grassland were closely associated with population growth in western China.

Based on demographic, socioeconomic, and policy considerations, we believe that spontaneous immigration eventually thoroughly altered the nomadic economy of this region into an agricultural one and resulted in the large-scale conversion of grasslands to cultivated land (Line 343-376).

Point 15: 359-361 the mentioned references do not support what was said in the text, and the remainder of the paragraph is not supported with a single reference. The use of existing literature needs to be much clearer and consistent, but also pertinent to the topic.

Response 15: Following this comment, we rewrote the paragraph and paid more attention to related references (Line 389-393).

Specifically, there has been such a high proportion of animal husbandry and a large area of pasture in Europe and the United States over a long period of time especially in the Alps of central Europe and the western mountain areas of the United States [1]. Land use strategy in these regions is that population growth stimulates the development of animal husbandry and therefore enlarges pasture areas [2-4].

References

Ramankutty, N.; Evan, A.T.; Monfreda, C.; Foley, J.A. Farming the planet: 1. Geographic distribution of global agricultural lands in the year 2000. Glob. Biogeochem. Cycle 2008, 22, GB1003. Klein Goldewijk, K.; Beusen, A.; Doelman, J.; Stehfest, E. Anthropogenic land use estimates for the Holocene - HYDE 3.2. Earth Syst. Sci. Data 2017, 9, 927–953. Pongratz, J.; Reick, C.; Raddatz, T.; Claussen, M. A reconstruction of global agricultural areas and land cover for the last millennium. Glob. Biogeochem. Cycle 2008, 22, GB3018. Kaplan, J.O.; Ruddiman, W.F.; Crucifix, M.C.; Oldfield, F.A.; Krumhardt, K.M.; Ellis, E.C.; Ruddiman, W.F.; Lemmen, C.; Goldewijk, K.K. Holocene carbon emissions as a result of anthropogenic land cover change. The Holocene 2011, 21, 775–791.

Point 16: Line 382-386 discuss the definition of grasslands in this research. This would probably be better in introduction or methods section.

Response 16: Following this comment, we added the description on grassland definition in section “2. Materials and Methods” (Line 134-136). According to definitions, grassland refers to a kind of land cover; pasture is a kind of anthropogenic land use. Before this study, grassland cover data with a large area and a long time period were lacked in China. Global datasets such as PJ and SAGE contained long-time grassland data in China, but estimates for grasslands in them vary widely. The difference in definitions is one of the main reasons for the stark difference in the results between them. So, we clarify this issue in Discussion section.

Point 17: Chapter 4.2. needs to present the main findings in a clear manner. What is the main reason for the stark difference in the results presented in Figure 9?

Response 17: Following this comment, we tried our best to revise our expression in a clear manner. (Line 383-434, the red font refers to that our expression has been modified, and the blue font represents the adjustment of its position).

Land use practices, the definition, and the reconstruction method are the main reason for the difference between them.

Point 18: Generally, 4.2., 4.3. and 4.4. are an interesting contribution, but it was not one of the aims or objectives of the paper, so it is not clear how it fits in the paper overall. You need to be clear at what you are trying to present and why. Is the aim to find trends in grassland change? Or is it to assess the drivers of change? Or is it to compare the quality of different global datasets and assess it to local/regional level?

Response 18: This study aims to reconstruct historical grassland cover change.

1) Before this study, global datasets such as PJ and SAGE contained long-time grassland data in China. But historical data and our current understanding do not support the results of global datasets. Therefore, it is necessary to make a comparison between them from definition, method to final result (section 4.2). Following this comment, we revised our expression (Line 383-434).

2) Modern LUCC information can be obtained through monitoring of satellite remote sensing, aerial photographs, and field surveys; past LULCC is, however, limited in space and time due to the lack of direct, large-scale observations. To reconstruct historical grassland, we set some reasonable assumptions and then propose a past LULCC scenario based on the relevant knowledge of modern LULCC. But the scenario does not equal the facts and has still existed many shortcomings. Therefore, we point out the shortcomings of reconstruction methods and future directions for improvement (section 4.3).

3) In Introduction section, we emphasized reconstructing historical grassland cover change can provide significant basic data for global and regional environmental change modeling. We derived the grassland cover between 1661 and 1996 in western China. Then, to enhance our understanding of environmental change, we introduced what we could do using the results here in the future (section 4.4).

Point 19: Conclusion looks like results – too specific repetition of the actual results. Most important findings need to be reiterated generally and put into wider context.

Response 19: Following this comment, we rewrote section “5. Conclusions” (Line 485-501).

Specifically, based on grassland definition and land use practice in western China, this study presents a reconstruction-based model for historical grassland cover using information from PENG and CHCD. Grassland cover maps for western China were generated at a 10 km resolution for the period between 1661 and 1996. Results suggest that the natural grassland area across western China (without land reclamation) was originally 320.72 × 106 ha; models imply that decreasing trends were also observed in all western provinces from 1661 to 1996.

Grassland cultivation mainly occurred within the Gan-Ning area and was driven by agricultural development policies of the Qing government between 1661 and 1724. Agriculture was subsequently expanded in Inner Mongolia and northeastern China due to an increased number of refugees; this led to a slight decline in grassland area between 1724 and 1873. Over the period between 1873 and 1980, opening up of grasslands was mainly concentrated in northeastern China and Xinjiang. Large increases in both the intensity and extent of reclaimed grassland resulted from agricultural economic crisis, policies, and exponentially increased immigration.

Definitions for pasture, land use practice, and the reconstruction methods applied by global datasets like HYDE 3.2 and PJ cannot be considered suitable for western China. Data from previous compilations fail to objectively reveal the influence of long-term anthropogenic activities on grassland vegetation of western China.

Round 2

Reviewer 1 Report

the revisions are not excellent but sufficient

Author Response

Response: Thank you for your comment. Based on the comments and suggestions of reviewer 2, we revised our manuscript further.

Reviewer 2 Report

This is generally a good, coherent paper. You made a lot of necessary adjustments, even if some could still improve the manuscript (see comments below). My only issue is that the paper is still in very local, meaning that you did not make any links in the discussion with similar research outside China.

ABSTRACT

First three sentences in the abstract can be rewritten and shortened into one sentence.

"The reconstruction results enable clear reflect grassland cover changes over time and provide  crucial data that can be used for modeling climate change and carbon emissions." - The first part of the sentence doesn't make any sense, and the conclusions should be a bit more specific in this case.

Line 42 - what is "ecological" environment?
Line 47-48 - historical LULCC studies have indeed been often descriptive and without too much  quantitative analyisis in the past, but to claim that they have TO DATE been qualitative and statistically insignificant is not correct.

Line 90 - is it important for the scholars to be Chinese?
Line 96 - what do you mean when you say progress on historical cropland?
Sources for figure 1?
169-178 should be moved to study area and linked to methods there, not in the methods section.

Author Response

Manuscript ID: ijerph-570233

Title: Exploring spatiotemporal pattern of grassland cover in Western China from 1661 to 1996 using remote sensing and cropland data

Response to Reviewer 2 Comments

Comments and Suggestions for Authors:

This is generally a good, coherent paper. You made a lot of necessary adjustments, even if some could still improve the manuscript (see comments below). My only issue is that the paper is still in very local, meaning that you did not make any links in the discussion with similar research outside China.

Response: Thank you for this comment. We revised the Abstract (Line 14-33), Introduction (Line 37-103), Materials and Methods (Line 104-249), and Discussion (Line 322-489).

In Discussion section, we added more information and references to clarify the relevance between this study and similar studies outside China. “Currently, numerous regional grassland cover has been reconstructed outside China [1-4]. Generally, the research thoughts of different regions are consistent, that is, a large amount of grassland was converted into cropland and pasture due to population growth. But, there are obvious regional differences in reconstruction methods because of different anthropogenic land use practices. It is therefore not feasible to reconstruct historical grassland cover with a uniform method on global scale; conversely, the reconstruction results in this way are uncertain at regional scale, such as HYDE 3.2 and PJ. The idea of dividing different regions, time sections and fine categories can be adopted to reconstruct spatial pattern of historical grassland.” (Line 433-440).

References

Fuchs, R.; Herold, M.; Verburg, P.H.; Clevers, J.G.P.W. A high-resolution and harmonized model approach for reconstructing and analysing historic land changes in Europe. Biogeosciences 2013, 10, 1543-1559. Yu, Z.; Lu, C. Historical cropland expansion and abandonment in the continental U.S. during 1850 to 2016. Glob. Ecol. Biogeogr. 2017, 27, 322-333. Cousins, S.A.O. Analysis of land-cover transitions based on 17th and 18th century cadastral maps and aerial photographs. Landsc. Ecol. 2001, 16, 41–54. Fuchs, R.; Verburg, P.H.; Clevers, J.G.P.W.; Herold, M. The potential of old maps and encyclopaedias for reconstructing historic European land cover/use change. Appl. Geogr. 2015, 59, 43–55.

(Note: the red font refers to that our expression has been modified in the revised manuscript, and the blue font represents the adjustment of its position.)

ABSTRACT

Point 1: First three sentences in the abstract can be rewritten and shortened into one sentence.

Response 1: Following this comment, we rewrote these sentences “Historical grassland cover change is vital for global and regional environmental change modeling, however, in China, its estimates are rare, and therefore, we proposed a method to reconstruct grassland cover over the past 300 years.” (Line 14-16).

Point 2: "The reconstruction results enable clear reflect grassland cover changes over time and provide crucial data that can be used for modeling climate change and carbon emissions." - The first part of the sentence doesn't make any sense, and the conclusions should be a bit more specific in this case.

Response 2: Thank you for this suggestion. Following this comment, we revised the sentence “The reconstruction results enable provide crucial data that can be used for modeling long-term climate change and carbon emissions.” (Line 32-33).

Point 3: Line 42 - what is "ecological" environment?

Response 3: Following this comment, we revised the expression “serious degradation of the ecological environment” to “serious degradation of terrestrial ecosystems”. (Line 40).

Point 4: Line 47-48 - historical LULCC studies have indeed been often descriptive and without too much quantitative analysis in the past, but to claim that they have TO DATE been qualitative and statistically insignificant is not correct.

Response 4: Thank you. Following this comment, we deleted the incorrect phrase (Line 45-46).

Point 5: Line 90 - is it important for the scholars to be Chinese?

Response 5: Following this comment, we revised the expression “A few Chinese scholars” to “A few scholars”. (Line 87).

Point 6: Line 96 - what do you mean when you say progress on historical cropland?

Response 6: Progress refers to that scholars have reconstructed Chinese croplands for the last millennium [1-5] and forests for the last 300 years [6-8]. But, historical grassland reconstructions have been limited to just single periods or regions. Also, we revised our expression “Compared with progresses on reconstructing historical croplands and forests in China”. (Line 92-93).

References

Li, M.J.; He, F.N.; Yang, F.; Li, S.C. Reconstructing provincial cropland area in eastern China during the early Yuan Dynasty(AD1271–1294). J. Geogr. Sci. 2018, 28, 1994–2006. Ye, Y.; Fang, X.Q.; Ren, Y.Y.; Zhang, X.Z.; Chen, L. Cropland cover change in Northeast China during the past 300 years. Sci. China-Earth Sci. 2009, 52, 1172–1182. Li, S.C.; He, F.N.; Zhang, X.Z. A spatially explicit reconstruction of cropland cover in China from 1661 to 1996. Reg. Envir. Chang. 2016, 16, 417–428. He, F.N.; Li, M.J.; Li, S.C. Reconstruction of Lu-level cropland areas in the Northern Song Dynasty (AD976–1078). J. Geogr. Sci. 2017, 27, 606–618. Li, M.J.; He, F.N.; Li, S.C.; Yang, F. Reconstruction of the cropland cover changes in eastern China between the 10th century and 13th century using historical documents. Sci Rep 2018, 8, 13552. He, F.N.; Ge, Q.S.; Dai, J.H.; Rao, Y.J. Forest change of China in recent 300 years. J. Geogr. Sci. 2008, 18, 59–72. He, F.N.; Li, S.C.; Zhang, X.Z. A spatially explicit reconstruction of forest cover in China over 1700–2000. Glob. Planet. Change 2015, 131, 73–81. Yang, X.H.; Jin, X.B.; Xiang, X.M.; Fan, Y.T.; Shan, W.; Zhou, Y.K. Reconstructing the spatial pattern of historical forest land in China in the past 300 years. Glob. Planet. Change 2018, 165, 173–185.

Point 7: Sources for figure 1?

Response 7: The digital elevation model (DEM) was obtained from Geospatial Data Cloud (http://www.gscloud.cn/). And we added more information on the source of figure 1 (Line 124-125).

Point 8: 169-178 should be moved to study area and linked to methods there, not in the methods section.

Response 8: Thank you for this suggestion. Following this comment, we moved Line 169-178 to section “2.1. Study area”. (Line 126-134).